# Evidence of centromeric histone 3 chaperone involved in DNA damage repair pathway in budding yeast

Prakhar Agarwal, Anushka Alekar, Shubhomita Mallick, Kannan Harini, Santanu K Ghosh*

Department of Biosciences and Bioengineering, Indian Institute of Technology Bombay, Mumbai, India

**Abstract** The centromeric protein-A (CENP-A) is an evolutionarily conserved histone H3 variant that marks the identity of the centromeres. Several mechanisms regulate the centromeric deposition of CENP-A as its mislocalization causes erroneous chromosome segregation, leading to aneuploidy-based diseases, including cancers. The most crucial deposition factor is a CENP-A specific chaperone, HJURP (Scm3 in budding yeast), which specifically binds to CENP-A. However, the discovery of HJURP as a DDR (DNA damage repair) protein and evidence of its binding to Holliday junctions in vitro indicate a CENP-A-deposition-independent role of these chaperones. In this study, using budding yeast, we demonstrate that Scm3 is crucial for the DDR pathway as Scm3-depleted cells are sensitive to DNA damage. We further observe that Scm3 depletion genetically interacts with the *rad52* DDR mutant and is compromised in activating DDR-mediated arrest. We demonstrate that Scm3 associates with DNA damage sites and undergoes posttranslational modifications upon DNA damage. Overall, from this report and earlier studies on HJURP, we conclude that DDR functions of CENP-A chaperones are conserved across eukaryotes. The revelation that these chaperones promote genome stability in more than one pathway has clinical significance.

*For correspondence:
santanughosh@iitb.ac.in

Competing interest: The authors declare that no competing interests exist.

## Editor's evaluation

DNA breaks occur frequently at centromeres and their repair is essential to maintain mitotic integrity in eukaryotes. The authors address this important question using budding yeast as a model system, with convincing data that demonstrates a role for the centromeric histone H3 chaperone Scm3 contributing to this repair pathway. These data have exciting implications for repair pathways that regulate an essential chromosomal locus required for mitosis and cell viability. The paper will be of interest to the centromere, chromosome instability and DNA repair fields.

## Introduction

The faithful replication and partitioning of the replicated chromatin as chromosomal units from mother to daughter cell is important for cell survival. New nucleosomes are assembled on DNA during replication, which requires the deposition of histones in a highly regulated step-by-step process that involves the deposition of an $(H3-H4)_2$ tetramer followed by two H2A-H2B dimers (*Hammond et al., 2017*; *Ransom et al., 2010*). The deposition of histones is assisted by a group of proteins called histone chaperones. These chaperones physically interact with the histones in a pre-nucleosomal complex and thus prevent premature histone-DNA interaction or nucleosome assembly (*Burgess and Zhang, 2013*; *Ransom et al., 2010*). The deposition of the canonical histones occurs mainly during DNA replication via a replication-coupled (RC) pathway using chaperones dedicated to the RC pathway

(*Serra-Cardona and Zhang, 2018*). For instance, Asf1 and CAF-1 deposit (H3-H4)$_2$ tetramers (*Burgess and Zhang, 2013*; *De Koning et al., 2007*) and Nap1 facilitates (H2A-H2B)$_2$ deposition (*De Koning et al., 2007*; *Eitoku et al., 2008*) into nucleosomes. On the other hand, variant histones that earmark specialized chromatin domains are deposited at specific sites throughout the cell cycle using the replication-independent (RI) pathway using variant-specific chaperones dedicated to the RI pathway (*Smith, 2002*). For instance, deposition of the variant histones H2A.Z, CENP-A, and MacroH2A, important in transcriptional activation (*Gerton et al., 2000*), kinetochore formation (*Earnshaw and Rothfield, 1985*; *Earnshaw and Tomkiel, 1992*; *McKinley and Cheeseman, 2016*) and X chromosome inactivation (*Sun and Bernstein, 2019*), occurs via chromatin remodelers (*Fan et al., 2022*), the HJURP chaperone (*Dunleavy et al., 2009*; *Foltz et al., 2009*), and acidic nuclear phosphoprotein 32B (ANP32B; *Mandemaker et al., 2023*), respectively.

Almost all eukaryotes harbor CENP-A at centromeres, which promotes the formation of kinetochores at those loci (*Blower and Karpen, 2001*; *Buchwitz et al., 1999*; *Howman et al., 2000*; *Stoler et al., 1995*; *Van Hooser et al., 2001*; *Earnshaw and Rothfield, 1985*). In metazoans, HJURP (Holliday junction recognition protein) is the cognate chaperone (*Dunleavy et al., 2009*; *Foltz et al., 2006*) of CENP-A. The CENP-A binding domain at the N-terminus of HJURP is highly conserved (*Aravind et al., 2007*; *Sanchez-Pulido et al., 2009*; *Shuaib et al., 2010*). Despite the well-established function of HJURP in CENP-A deposition at centromeres, HJURP was initially identified as a protein involved in the DNA damage response (DDR; *Kato et al., 2007*). HJURP was so named because of its ability to bind in vitro to Holliday junction DNA substrates generated in vivo during DNA damage repair through homologous recombination (*Kato et al., 2007*). HJURP was found to interact physically with the DNA damage sensing complex MRN (Mre11-Rad50-Nbs1), and the interaction was found to increase significantly upon induction of double-strand breaks (DSBs) (*Kato et al., 2007*). As a corollary, the level and subcellular localization of HJURP change in cells treated with DNA-damaging agents such as ionizing radiation (IR), hydroxyurea (HU), cisplatin, and camptothecin (CPT; *Kato et al., 2007*). Finally, the expression of HJURP was found to be regulated by ATM, a DDR kinase, as no protein expression was observed in ATM-deficient cancerous cell lines. Notably, HJURP has been found to be diffusely present throughout the nucleus (*Dunleavy et al., 2009*; *Kato et al., 2007*), in agreement with its involvement in DDR.

Given the functions of HJURP in CENP-A deposition and DDR pathway, it is not surprising that misregulation of HJURP is involved in disease states. HJURP is considered an oncogene as its overexpression has been associated with cancers (*Chen et al., 2018*; *Kang et al., 2020*) including breast cancer (*Coates et al., 2010*; *Hu et al., 2010*; *Montes de Oca et al., 2015*), lung cancer (*Kato et al., 2007*), ovarian cancer (*Dou et al., 2022*), glioblastoma (*Valente et al., 2013*; *Valente et al., 2009*), hepatocellular carcinoma (*Chen et al., 2018*; *Hu et al., 2017*; *Luo et al., 2022*), colorectal cancer (*Kang et al., 2020*), and pancreatic cancer (*Wang et al., 2020*). Consequently, the inhibition of HJURP attenuates the cancer cells' properties and increases the survival rate of patients, and thus, HJURP acts as a prognostic marker for several cancers (*Kang et al., 2020*; *Lai et al., 2021*).

In budding yeast, Cse4 and Scm3 are the homologs of CENP-A and HJURP, respectively. Scm3 deposits Cse4 at centromeres during S phase through its physical interaction with Ndc10, which binds to centromeric DNA as part of the CBF3 complex (*Camahort et al., 2007*; *Cho and Harrison, 2012*; *Cho and Harrison, 2011*). Notably, following the deposition of Cse4, Scm3 remains associated with the centromeres throughout the cell cycle (*Wisniewski et al., 2014*; *Xiao et al., 2011*), and like HJURP, Scm3 has been found on non-centromeric chromatin (*Mizuguchi et al., 2007*). Besides the highly conserved CENP-A binding domain, both HJURP and Scm3 also harbor a homologous DNA binding domain (*Figure 1—figure supplement 1A*), which is predicted to be evolutionarily conserved in all vertebrates and fission yeast (*Müller et al., 2014*). Overexpression of Scm3, like HJURP, causes chromosome instability, and it is believed that Scm3, when in excess, binds to centromeric DNA independently and leads to chromosome missegregation (*Choy et al., 2012*; *Mishra et al., 2011*). Both Scm3 and HJURP form homo-dimers, and in an in vitro assay, Scm3 and HJURP bind to non-centromeric DNA (*Kato et al., 2007*; *Stoler et al., 2007*; *Xiao et al., 2011*). Non-homologous proteins, such as budding yeast Ndc10 and mammalian Mis18, are critical for Cse4 and CENP-A deposition, respectively, but they bind to Scm3 and HJURP through regions other than the conserved Cse4/CENP-A- and DNA-binding domain of these chaperones. Therefore, despite the significant difference in polypeptide length between HJURP and Scm3 (748 vs 229), their functions appear conserved from

humans to yeast. Given the physical interaction of HJURP with the MRN complex, it is highly plausible that Scm3 might also interact with DDR proteins in yeast. Altogether, from the above observations, we hypothesize that Scm3, like HJURP, may be involved in the DDR pathway in budding yeast. In this work, we report that cells lacking Scm3 are sensitive to DNA-damaging agents and are compromised in DNA damage repair. In addition to the known functions of HJURP in the DDR pathway, in this report, we show that Scm3 is required for mounting DDR-dependent checkpoint activation. Importantly, we demonstrate that Scm3 associates with DNA damage sites upon induction of DNA damage and persists there, perhaps to facilitate damage repair. Our results suggest that the phosphorylation of Scm3 might be involved in its DDR-related functions. Taken together, this study reveals an evolutionary kinship amongst CENP-A chaperones in conferring genome stability through efficient DNA damage repair, besides their canonical role in CENP-A deposition.

## Results

### Scm3-depleted cells are sensitive to MMS-mediated DNA damage

Yeast cells compromised in DNA damage repair (DDR) show sensitivity to DNA-damaging agents such as methyl methane sulfonate (MMS), HU, and CPT. To investigate the role of Scm3 in DDR, we tagged Scm3 with an auxin-inducible degron to conditionally suppress Scm3 function (*SCM3-AID*; *Nishimura et al., 2009*) since the protein is essential for growth (*Camahort et al., 2007*; *Mizuguchi et al., 2007*; *Stoler et al., 2007*), and assayed the behavior of Scm3-depleted cells in the presence of DNA damaging agents. Since cells devoid of the homologous recombination protein, Rad52, are known to show extreme sensitivity to such drugs, *rad52Δ* cells were used as a positive control in our assays (*Plate et al., 2008*; *Kaytor and Livingston, 1994*). Mid-log grown cells were harvested, serially diluted, and spotted on plates supplemented with auxin and DNA damaging agents (MMS, HU, CPT) in different combinations (*Figure 1A*, *Figure 1—figure supplement 2A*). We observed that *SCM3-AID* cells (three independent transformants) were sensitive to MMS in the presence but not the absence of auxin. *rad52Δ* cells, as expected, were sensitive to MMS but not to auxin (*Figure 1A*). To quantify the levels of sensitivity towards MMS, we performed a colony-forming unit (CFU) count assay where *SCM3-AID* cells were pre-grown in rich media without auxin to mid-log phase before an equal number of cells was plated on media supplemented with auxin and MMS in different combinations (*Figure 1B*). The number of colonies formed after 36 hr was counted, and the number of colonies on plates free of auxin and MMS was interpreted as 100% cell viability. We observed a statistically significant drop in cell viability on the plate supplemented with both auxin and MMS compared to auxin alone (*Figure 1B*).

Since the presence of auxin alone suppresses the growth of *SCM3-AID* cells, we took yet another approach to examine the sensitivity of these cells to MMS. We pre-treated the cells with 0.75 mM auxin for 2 hr to deplete Scm3, then washed out auxin and spotted on plates containing different concentrations of MMS, without auxin. The depletion of Scm3 was verified by observing a higher percentage of G2/M arrested cells and by Western blot verifying the degradation of Scm3-AID after auxin treatment (*Figure 1—figure supplement 2B–C*). Here also, we found that Scm3-depleted cells were more sensitive to MMS as compared to the wild-type cells (*Figure 1—figure supplement 2D*). To quantify the sensitivity observed by this approach, we performed a CFU count, as mentioned above. Following auxin pretreatment, an equal number of wild-type, *SCM3-AID,* and *rad52Δ* cells were spread on plates with or without 0.01% MMS. We again found a statistically significant drop in cell viability on the MMS plate when Scm3 was pre-depleted (*Figure 1—figure supplement 2E*). The *rad52Δ* cells, as the positive control, showed high sensitivity to MMS, as expected.

The increased sensitivity of Scm3-depleted cells to DNA-damaging agents could be due to the weakening of the kinetochores, as Scm3-mediated deposition of Cse4 promotes kinetochore assembly, or due to the delay in the cell cycle, as Scm3-depleted cells arrest in late G2/M phase (*Camahort et al., 2007*; *Cho and Harrison, 2011*). If either of these hold true, perturbation of the kinetochore by degradation of other kinetochore proteins or wild-type cells arrested at metaphase would likely show a similar sensitivity to MMS. In budding yeast, Ndc10 is recruited to the centromeres upstream of Scm3 (*Lang et al., 2018*), whereas the centromeric localization of Mif2, another essential inner kinetochore protein, depends on Scm3 and Cse4 (*Xiao et al., 2017*). We constructed *NDC10-AID* and *MIF2-AID* strains and used them for our assay to represent proteins independent or dependent

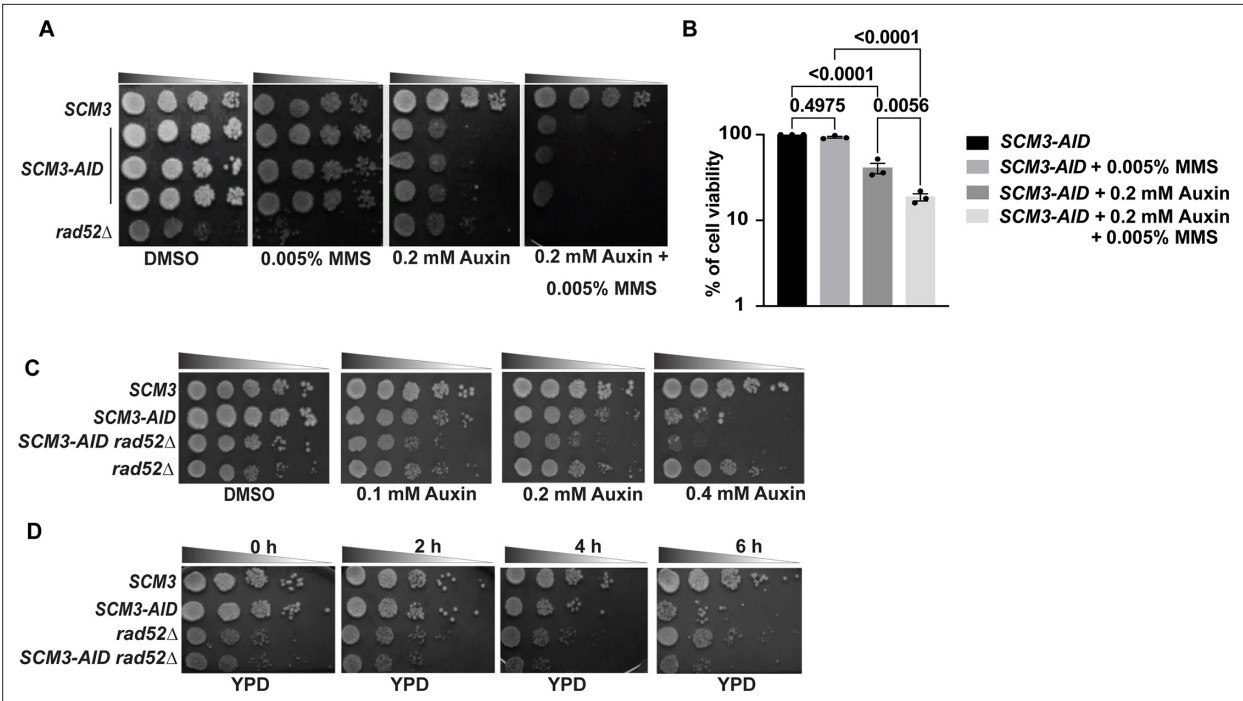

**Figure 1.** Depletion of Scm3 causes sensitivity to MMS. (**A**) Wild-type (*SCM3*), *SCM3-AID* (three independent transformants), *and rad52Δ* cells, grown till mid-log, were serially diluted and spotted on the indicated plates. The plates were incubated at 30 °C for 24–48 hr before imaging. (**B**) An equal number of mid-log grown *SCM3-AID* cells were spread on plates containing the indicated supplements. Cell viability was estimated from the number of colonies that appeared after incubation at 30 °C for 2 days (see Materials and methods). The colony number on the plate without auxin and MMS was taken as 100% viability. Error bars represent the standard error of the mean. The p-values were estimated by a one-way ANOVA test followed by Tukey's multiple comparison test from three independent experiments. (**C**) Wild-type (*SCM3*), *SCM3-AID, SCM3-AID rad52Δ,* and *rad52Δ* cells, grown till mid-log, were serially diluted and spotted in the presence of the indicated auxin concentrations. The plates were incubated at 30 °C for 24–48 hr before imaging. (**D**) The cells of the strains used in (**C**) were grown till mid-log and treated with 0.75 mM auxin for the indicated duration before they were washed, serially diluted, and spotted on YPD plates that were incubated at 30 °C for 24–48 hr before imaging.

The online version of this article includes the following source data and figure supplement(s) for figure 1:

**Source data 1.** Source data for *Figure 1B*.

**Figure supplement 1.** NLS domain and the C-terminal amino acids of Scm3 are dispensable for the DDR function.

**Figure supplement 2.** Cells lacking Scm3 are sensitive to various DNA damaging agents.

**Figure supplement 2—source data 1.** Original files for western blot analysis displayed in *Figure 1—figure supplement 2C*.

**Figure supplement 2—source data 2.** Original files for western blot analysis displayed in *Figure 1—figure supplement 2C* indicating relevant bands.

**Figure supplement 2—source data 3.** Source data for *Figure 1—figure supplement 2E*.

**Figure supplement 3.** MMS sensitivity of the cells lacking Scm3 is independent of its kinetochore function or effect on cell cycle progression.

on Scm3 for centromeric localization, respectively. We also included one non-essential kinetochore protein, Ctf19, a protein of the COMA complex, to remove any possible misjudgment in distinguishing a cell-growth-arrest phenotype occurring due to drug sensitivity vs. auxin-mediated degradation of essential proteins. The COMA complex is directly recruited to the centromeres through interaction with the N-terminal tail of Cse4, hence dependent on Scm3 (*Chen et al., 2000*; *Fischböck-Halwachs et al., 2019*). Mid-log phase cells were harvested and spotted on plates with or without MMS and/or auxin; however, we did not observe any increased sensitivity of such cells to MMS (*Figure 1—figure supplement 3*). Further, wild-type cells, when challenged in the presence of nocodazole and MMS, also did not show an increased sensitivity to MMS. Therefore, the increased sensitivity to MMS in Scm3-depleted cells but not in other kinetochore mutant or metaphase-arrested cells indicates that Scm3 possesses an additional function in genome stability besides its role in kinetochore assembly.

Since HJURP is involved in DDR through the HR pathway (*Kato et al., 2007*), Scm3-depleted cells are expected to interact genetically with HR mutants if Scm3 shares this function with HJURP. We therefore tested genetic interaction between Scm3-depleted and *rad52Δ* cells, as Rad52 is essential

for efficient homologous recombination in *S. cerevisiae* (**Barlow and Rothstein, 2010**; **Lee et al., 2003**). In the presence of auxin, the double mutant grew slowly compared to the single mutants and the wild-type strain (**Figure 1C**). To further validate the genetic interaction between the two genes, mid-log grown cells were pre-treated with 0.75 mM auxin to deplete Scm3 before cells were spotted on YPD plates. Again, we observed that the double mutant grew slower than the wild type and single mutants (**Figure 1D**). Next, we wanted to investigate how the double mutant behaves when challenged by a DNA damaging agent. As expected, when the double mutant was spotted in the presence of 0.005% MMS, the single mutants grew better than the double mutant (**Figure 1—figure supplement 2F**). These results indicate a genetic interaction between Scm3 depletion and *rad52* deletion.

Further, we wanted to identify the regions in Scm3 whose loss confers sensitivity to DNA damage. Scm3 contains two essential conserved regions, NES (nuclear export signal) and HR (heptad repeat), having functions in DNA and Cse4 binding, respectively (**Figure 1—figure supplement 1A**). Scm3 also has a non-essential D/E-rich region, which is important for protein stability but not involved in Cse4 binding (**Stoler et al., 2007**). In addition, Scm3 possesses two BR (bromodomain) regions, each harboring a candidate NLS (nuclear localization signal). Deletion of the C-terminal 25 amino acids or the bromodomain regions does not cause cell lethality (**Shivaraju et al., 2011**; **Stoler et al., 2007**). We used these non-essential *scm3* mutants (**Shivaraju et al., 2011**) as the sole source of Scm3 and challenged the cells to MMS and CPT along with wild-type Scm3-expressing cells as a control. We found no growth defect in response to MMS or CPT (**Figure 1—figure supplement 1B**), suggesting that at least the C-terminal 25 amino acids and bromodomain regions are not important for the DDR function of Scm3. This indicates that the essential NES and/or HR might be involved, but this is technically difficult to test using growth-based assays.

From these results, we conclude that the loss of Scm3 confers sensitivity to DNA-damaging agents, indicating that Scm3, like its mammalian homolog HJURP, may play a role in the DDR pathway.

## The loss of Scm3 generates more Rad52 foci

Having shown that the Scm3-depleted cells are sensitive to DNA damage, we hypothesized that these cells might be deficient in repairing DNA damage. To examine this, we wished to visualize Rad52-GFP foci in these cells. In budding yeast, the primary pathway to repair double-strand breaks is through homologous recombination (HR) mediated by Rad52. Therefore, loss of *RAD52* causes these DNA breaks to accumulate upon damage (**Dornfeld and Livingston, 1991**; **Lisby et al., 2003**; **Lisby et al., 2001**). Mid-log phase *RAD52-GFP SCM3-AID* cells were treated with auxin or left untreated for 2 hr. The cultures were then split and incubated with or without 0.02% MMS for 90 min (**Figure 2A**), followed by visualization of Rad52-GFP foci (**Figure 2B**). Interestingly, we observed that 13% of Scm3-depleted (+auxin) cells showed Rad52-GFP foci in the absence of MMS, in contrast to only 5% of wild-type cells (**Figure 2C**, untreated). Consistent with this observation, when these cells were challenged with MMS, the number of Rad52-GFP foci increased, and Scm3-depleted cells still showed Rad52-GFP foci more frequently (70%) than wild-type cells (45%) (**Figure 2C**, 0.02% MMS). Moreover, while the Scm3-depleted cells harbored an average of ~3 foci in MMS, the wild type showed ~2 foci per cell (**Figure 2D**). To rule out the possibility that auxin treatment alone can cause increased Rad52-GFP foci formation, we challenged the wild-type (*RAD52-GFP*) cells with auxin or DMSO and counted the number of cells with Rad52-GFP foci. We did not observe an increase in Rad52-GFP-positive cells when treated with auxin +DMSO, as compared to only DMSO (**Figure 2—figure supplement 1A–B**). We further quantified the distribution of cells with 1, 2, 3, or >4 Rad52-GFP foci in *SCM3-AID* cells treated with MMS with or without auxin. Scm3-depleted cells showed a significantly higher number of cells with 3 or >4 Rad52-GFP foci (**Figure 2—figure supplement 1C**). These results demonstrate that loss of Scm3 results in either an increased number of DSBs or a change in repair kinetics.

To understand the dynamics of Rad52 foci formation in Scm3-depleted cells, we used an alpha factor arrest and release (**Figure 2—figure supplement 2A**). Briefly, *bar1Δ RAD52-GFP SCM3-AID* cells were arrested in the G1 phase of the cell cycle by adding alpha factor. 1 hr before release from G1 phase, cells were treated with auxin to deplete Scm3 or with the solvent DMSO as control. Subsequently, cells were washed and released into alpha factor-free media with or without MMS with either DMSO or auxin and with nocodazole to have all cells eventually arrest at metaphase. Following the alpha factor release, cells were harvested every 30 min and the percentage of cells containing

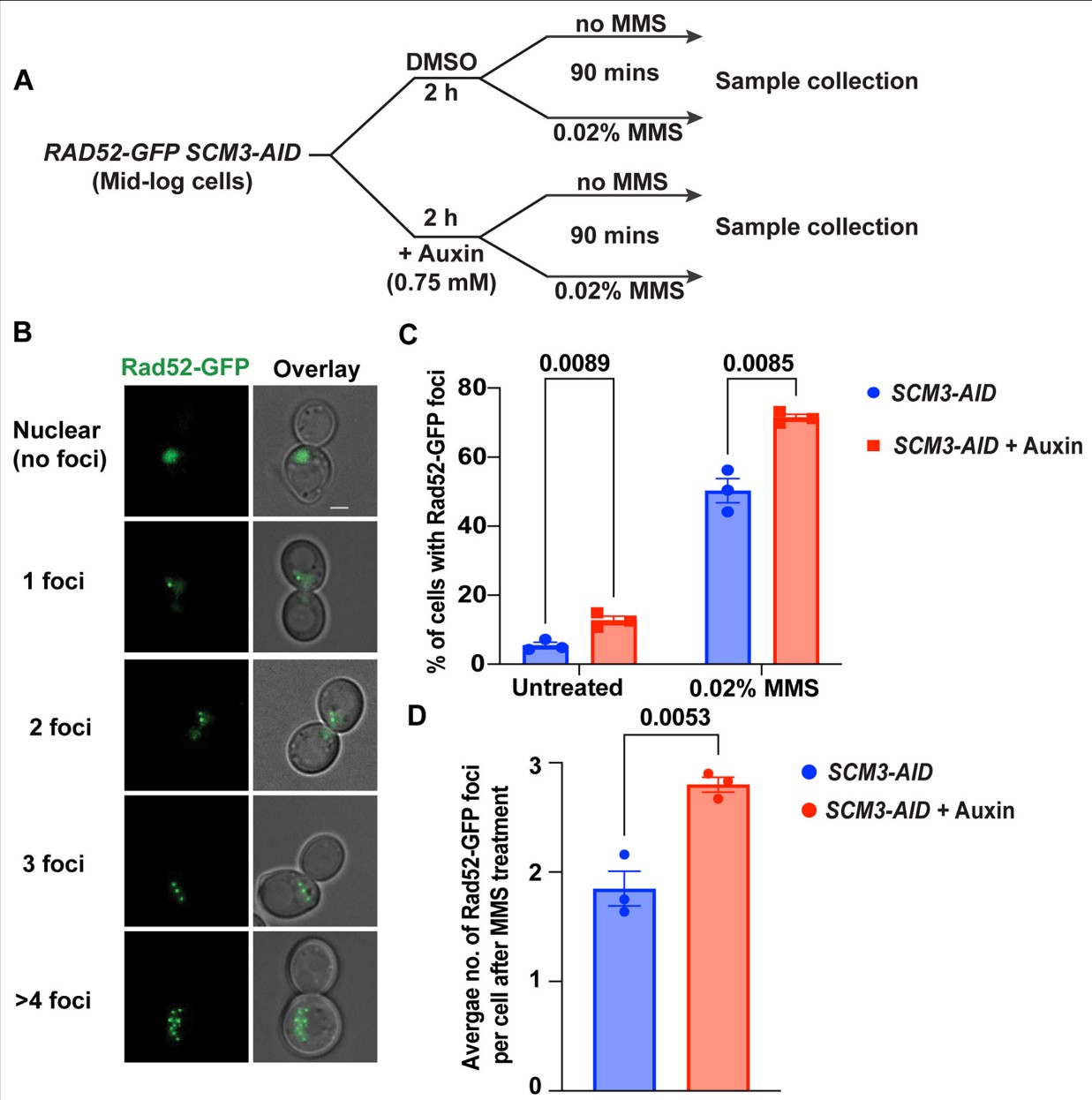

**Figure 2.** Depletion of Scm3 causes more Rad52 foci in the presence or absence of MMS. (**A**) Schematic of the experimental strategy for visualization of Rad52-GFP foci in the *RAD52-GFP SCM3-AID* cells either in the presence of DMSO or auxin (Scm3 depleted) and each either treated with MMS or left untreated. (**B**) Representative images showing different patterns of Rad52-GFP foci. Rad52-GFP produces a nuclear signal (row 1) in the absence of DNA damage, but with damage coalesces to form bright foci (rows 2–5) representing the DNA repair sites. (**C**) The percentage of indicated cells containing Rad52-GFP foci with or without MMS treatment for 90 min. (**D**) The average number of Rad52-GFP foci per MMS-treated cell from the experiment described in (**C**). At least 300 cells were analyzed from three independent experiments for each set in (**C**) and (**D**). Error bars represent the standard error of the mean. The p-values were estimated by an unpaired two-tailed Student's t-test. Scale bar = 2 μm.

The online version of this article includes the following source data and figure supplement(s) for figure 2:

**Source data 1.** Source data for *Figure 2*.

**Figure supplement 1.** Depletion of Scm3 leads to the formation of multiple DSB (Rad52-GFP) foci.

**Figure supplement 1—source data 1.** Source data for *Figure 2—figure supplement 1*.

**Figure supplement 2.** Depletion of Scm3 causes increased Rad52 foci along the cell cycle and cell death.

**Figure supplement 2—source data 1.** Source data for *Figure 2—figure supplement 2*.

Rad52-GFP foci was determined (*Figure 2—figure supplement 2B*). Without MMS, Scm3-depleted cells, but not the wild type, showed a peak of Rad52-GFP accumulation at around 60 min after G1 release (*Figure 2—figure supplement 2C*). Notably, this time point coincides with the timing of DNA replication (judged by bud morphology) when DNA is most susceptible to damage. In the presence of MMS, both cells with or without auxin showed a peak of Rad52-GFP accumulation at 60 min after release, but the fraction of cells with Rad52-GFP foci was higher when auxin was present, that is Scm3 depleted (*Figure 2—figure supplement 2D*). Taken together, Scm3-depleted cells exhibit more Rad52 foci, indicating a compromised DDR pathway in these cells. However, defects in DNA replication or the creation of other DNA lesions producing more foci cannot be ruled out.

If the DNA lesions remain unrepaired, the cells might undergo an apoptotic pathway leading to cell death. To determine the percentage of cell death, we conducted an experiment as outlined in *Figure 2A* and stained with propidium iodide (PI), which does not permeate live cells and stains only dead cells (*Figure 2—figure supplement 2E*, arrow). We observed that ~15% of the Scm3-depleted (+auxin, -MMS) cells were PI positive, in contrast to only ~4% for the wild type (DMSO, -MMS; *Figure 2—figure supplement 2E*, untreated). The PI-positive cell population increased after MMS treatment in both cultures, with a significant difference between them (35% vs 27%; *Figure 2—figure supplement 2*, 0.02% MMS). These results suggest that DNA damage in the absence of Scm3 leads to compromised cell viability.

## Scm3 associates with non-centromeric sites in response to DNA damage

Having observed that Scm3-depleted cells accumulate DNA damage and that Scm3-depleted cells interact genetically with *rad52D*, we hypothesized that Scm3 may be recruited to DNA damage sites. To examine this, we first wanted to see if Scm3 alters its localization pattern in response to MMS-mediated DNA damage. Since Scm3 is also expected to localize at the kinetochores (*Mizuguchi et al., 2007*), we visualized Scm3 along with the kinetochore marker, Ndc10, on chromatin spreads. Cells harboring *NDC10-6HA SCM3-13MYC* were mock-treated or treated with 0.02% MMS for 90 min. Ndc10 showed bright foci representing the kinetochore cluster, whereas Scm3 was localized to multiple locations across chromatin, including colocalizing with Ndc10 (*Figure 3A*, arrow). A similar localization of Scm3 throughout the nucleus has been observed earlier in live cells (*Luconi et al., 2011*; *Wisniewski et al., 2014*). We also observed a significant enrichment of Scm3 at the rDNA loops irrespective of the exogenous DNA damage (*Figure 3A*, arrowhead). The enrichment of Scm3 at the rDNA loops may reflect a role in maintaining the highly repetitive organization of rDNA. HJURP has also been shown to localize to rDNA arrays in mammalian systems (*Kato et al., 2007*). Additionally, upon quantifying the total intensity of Scm3-13Myc and Ndc10-6HA normalized against the background intensity, we observed a statistically significant increase in Scm3-13Myc, but not Ndc10-6HA intensity in response to MMS treatment (*Figure 3B and C*). No change in the intensity of Ndc10-6HA post-MMS treatment is consistent with the absence of MMS sensitivity in Ndc10-depleted cells (*Figure 1—figure supplement 3A*).

Having observed an increased chromatin localization of Scm3 in response to MMS on chromatin spreads, we wished to verify this by determining the genome-wide association of Scm3 using ChIP-Seq. To rule out any cell cycle differences that may influence Scm3 DNA binding, we first arrested the cells harboring *SCM3-13MYC* at metaphase with nocodazole. The arrested cells were further mock-treated or treated with 0.05% MMS for an additional 90 min before they were harvested for ChIP. Both the MMS-treated and untreated samples were similarly arrested as judged by DAPI staining (*Figure 3—figure supplement 1A*). The Scm3 ChIP-seq signal was normalized with the input signal, and genome-wide plots were generated to see Scm3 binding along the entire chromosomes. We also performed ChIP-seq from a no-tag control strain that provided the background signal from unspecific interactions with the antibody or beads. The Scm3 ChIP-seq signal was normalized with the input signal, and upon quantification from MACS2, we observed the enrichment (peaks) of Scm3 at all 16 centromeres in both untreated and MMS-treated samples (*Figure 3D*, blue bars). Scm3 was also enriched at the 9.1 kb rDNA region on Chr XII in both untreated and treated samples (*Figure 3E*, at rDNA), consistent with our chromatin spread data (*Figure 3A*). Upon further analysis, the enrichment of Scm3 at rDNA, which harbors repetitive regions, was found to be ~1.4-fold in the treated cells (1.54-fold enrichment relative to input) compared to untreated cells

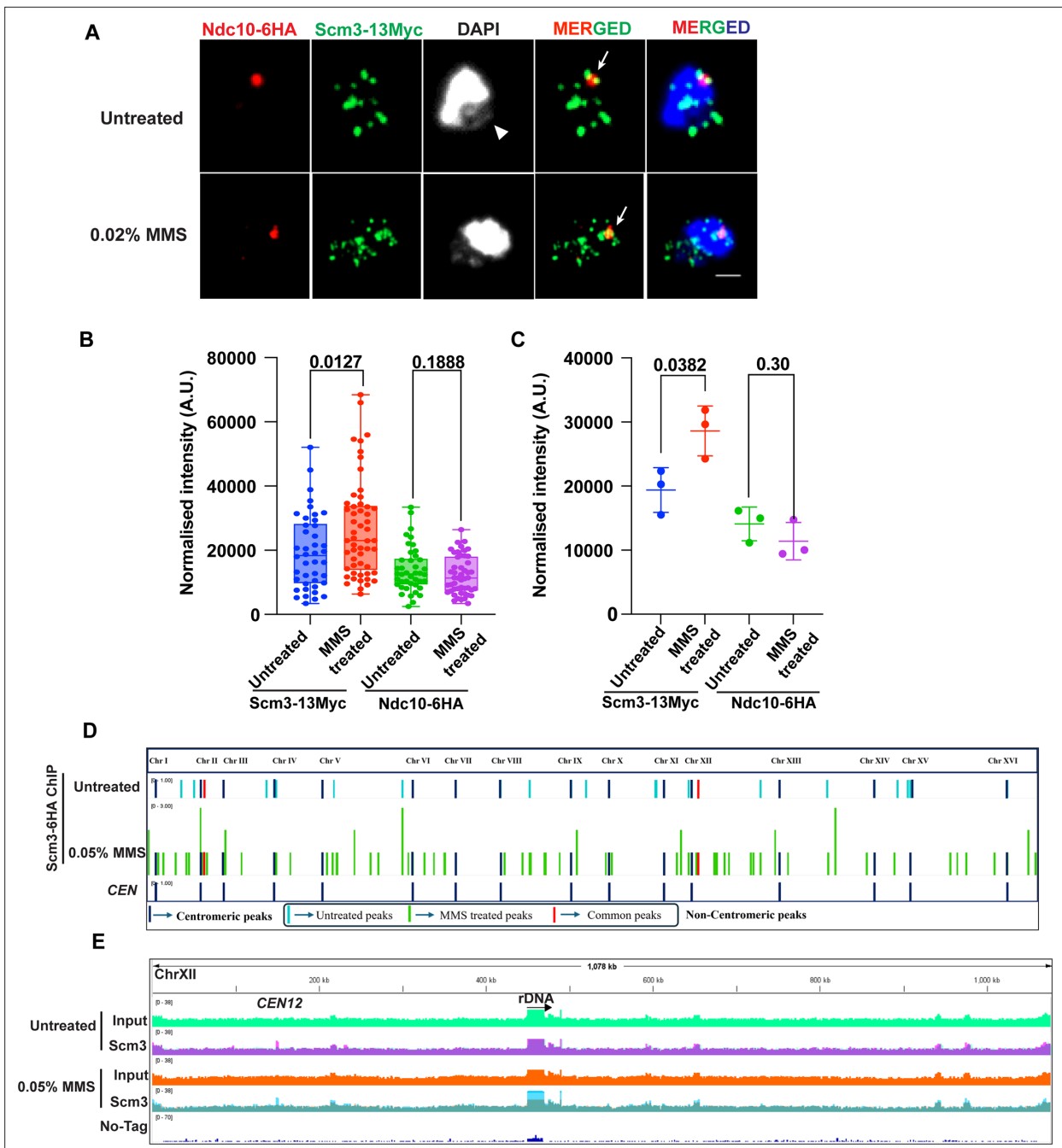

**Figure 3.** Localization of Scm3 in response to DNA damage. (**A**) Representative images showing the localizations of Ndc10-6HA and Scm3-13Myc using chromatin spreads. The mid-log grown *NDC10-6HA SCM3-13MYC* cells were treated with 0.02% MMS or mock-treated for 90 min prior to imaging. The arrows show the co-localization of Ndc10-6HA and Scm3-13Myc at the kinetochore cluster, and the arrowhead shows the rDNA loop. Scale bar = 2 µm. (**B**) The intensities of Scm3-13Myc and Ndc10-6HA after subtracting background intensity were quantified and plotted. (**C**) Mean values from three independent experiments. At least 30 spreads were analyzed for each set, as shown in (**B**). The p-values were determined by unpaired two-tailed student's t-test. (**D**) Whole genome view of the enriched peaks at the centromeric and non-centromeric regions in cells treated with 0.02% MMS or mock-treated. (**E**) Representative snapshot of chromosome XII, showing the peaks in the input and the samples (Scm3 IP). In the sample tracks (MMS-treated or mock-treated), the corresponding input signal is overlapped on the sample signal to clearly show the enrichment at the peaks.

The online version of this article includes the following source data and figure supplement(s) for figure 3:

**Source data 1.** Source data for *Figure 3*.

**Figure supplement 1.** Analysis of non-centromeric binding sites (peaks) of Scm3.

(1.06-fold enrichment relative to input; *Figure 3E*), suggesting its general preference towards such regions.

Besides the centromeric peaks, 17 non-centromeric peaks (~1.4–1.8-fold enrichment over input) were detected in untreated samples, compared to 75 peaks (~1.3–1.9-fold enrichment over input) in MMS-treated cells, in which only two peaks were shared (*Figure 3—figure supplement 1B*). This analysis clearly demonstrates an increase in the binding of Scm3 to the non-centromeric sites upon MMS treatment. The feature analysis of these sites revealed that MMS-treated samples were enriched in peaks which comprise fragile sites (*Song et al., 2014*) and recombination hotspots (*Gerton et al., 2000*), which are prone to DNA damage. The untreated samples had a comparatively smaller number of such damage-prone sites (*Figure 3—figure supplement 1C*, fragile sites and hotspots). Moreover, the MMS-treated samples had a higher number of peaks with short tandem repeats (length 5–7 nucleotides), interspersed repeats, and low complexity regions (less diversity of nucleotides) than in the untreated samples. These repeat regions are also shown to be a source of DNA damage, as they form alternative non-B DNA structures, which are more prone to damage (*Brown and Freudenreich, 2021*; *Wierdl et al., 1997*). Furthermore, analysis of the gene ontology (GO) associated with the 17 non-centromeric peaks revealed GO terms that are involved in normal metabolic pathways such as glycolysis/gluconeogenesis, regulation of amino acids, and telomere maintenance (*Figure 3—figure supplement 1E*) while in the treated samples the 75 non-centromeric peaks had GO terms that are mostly related to nucleotide synthesis and detoxification/flocculation, which are usually triggered in stress conditions (*Figure 3—figure supplement 1D*). We speculate that Scm3 might bind to these genes to ameliorate the DNA damage stress caused by MMS treatment. Hence, we could attribute the increased binding of the Scm3 to the non-centromeric regions to DNA damage.

To further understand the nature of the non-centromeric sites harboring the enriched peaks (17 untreated, 75 treated), they were scanned for the presence of any motifs, and we observed three and one significant motifs in the treated and in the untreated samples, respectively (*Figure 3—figure supplement 1F*). The three motifs in the MMS-treated samples were specifically observed in the viral LTR region of the gag-pol fusion gene, based on the annotation from the NCBI reference genome (NCBI accession: GCF_000146045.2). These viral LTRs are found in multiple chromosomes spanning 29 peaks (~39% of total peaks) in the treated samples. Interestingly, these regions are considered fragile sites marked by the binding of Rrm3 helicase (*Song et al., 2014*). Notably, these viral LTRs are also known to trigger DNA-damage response pathways (*Sinclair et al., 2006*). This suggests that upon DNA damage induction, Scm3 is recruited to genomic locations that are more prone to damage.

## Scm3 partially co-localizes with Rad52, perhaps at the DNA damage sites

Given the increased chromatin association of Scm3 in response to DNA damage, we wished to investigate if this occurs due to its binding to DNA damage sites. For this, we examined the colocalization of Scm3 with Rad52 upon DNA damage, as the latter is known to be targeted to the damage sites (*Lisby et al., 2003*; *Miyazaki et al., 2004*). Cells harboring *RAD52-6HA* and *SCM3-13MYC* were mock-treated or treated with 0.02% MMS for 90 min. We did not observe a significant co-localization between Scm3-13Myc and Rad52-6HA in untreated cells as judged by low Pearson's Correlation Coefficient value (PCC <0.4; *Figure 4A–C*). However, in the cells treated with MMS, we could observe a significant increase in the PCC value (PCC >0.45; *Figure 4B–C*). This indicates a partial co-localization between Scm3 and Rad52 in the presence of the DNA damage and implies that Scm3 may localize to the damage sites in response to DNA damage.

Next, we wished to examine the behavior of Scm3 with respect to another DNA damage marker. In mammals, the DSB sites are marked by γ-H2Ax, which also goes to other chromosomal locales, due to ATM/ATR kinases mediated phosphorylation of the H2A variant, H2Ax (*Burma et al., 2001*; *Lee et al., 2014*). In yeast, which lacks H2Ax, the DSBs are marked by γ-H2A due to phosphorylation of histone H2A by Mec1 and Tel1 kinases (*Lee et al., 2014*). Notably, γ-H2A is also found at the heterochromatin region and co-localizes with the yeast heterochromatin protein Sir3 (*Javaheri et al., 2006*; *Kirkland et al., 2015*; *Kitada et al., 2011*). We performed a chromatin spread assay using antibodies specific to γ-H2A and observed a drastic increase in the intensity of γ-H2A when the cells were subjected to DNA damage, which is expected (*Figure 4—figure supplement 1A–C*). To verify that the phenomenon is associated with DNA damage, we observed a significant increase in the colocalization between Rad52

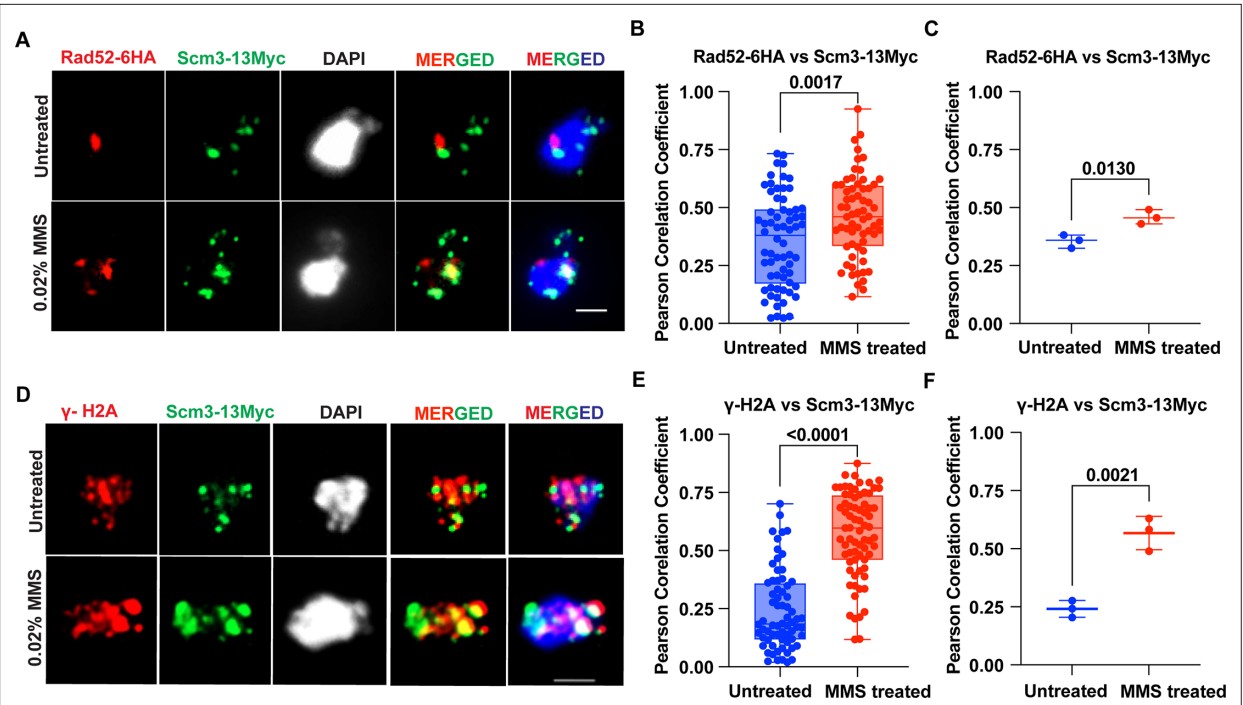

**Figure 4.** Scm3 colocalizes with DDR proteins in response to DNA damage. (**A**) Representative images showing the localizations of Rad52-6HA and Scm3-13Myc using chromatin spreads. The mid-log grown *RAD52-6HA SCM3-13MYC* cells were treated with 0.02% MMS or mock-treated for 90 min prior to imaging. (**B**) Quantification of the colocalization between Rad52-6HA and Scm3-13Myc using Pearson's correlation coefficient (PCC). (**C**) Mean values from three independent experiments. (**D**) Representative images showing the localizations of γ-H2A and Scm3-13Myc using chromatin spreads. The mid-log grown *SCM3-MYC* cells were treated with 0.02% MMS or mock-treated for 90 min prior to imaging. (**E**) Quantification of the colocalization between γ-H2A and Scm3-13Myc using PCC. (**F**) Mean values from three independent experiments. At least 30 spreads from three independent experiments were analyzed for each set in (**B**) and (**E**). The p-values were determined by unpaired two-tailed student's t-test. Scale bar = 2 μm.

The online version of this article includes the following source data and figure supplement(s) for figure 4:

**Source data 1.** Source data for *Figure 4*.

**Figure supplement 1.** Localization of proteins in presence or absence of DNA damage.

**Figure supplement 1—source data 1.** Source data for *Figure 4—figure supplement 1*.

**Figure supplement 2.** Hsf1 does not co-localize with γ-H2A in response to MMS or heat shock.

**Figure supplement 2—source data 1.** Source data for *Figure 4—figure supplement 2*.

and γ-H2A in response to DNA damage (*Figure 4—figure supplement 1D–F*). To further correlate the localization of Scm3 with DNA damage, we then visualized both Scm3-13Myc and γ-H2A simultaneously on chromatin spreads from *SCM3-13MYC* cells mock-treated or treated with 0.02% MMS for 90 min. We did not observe any significant co-localization between Scm3-13Myc and γ-H2A in untreated cells as judged by low Pearson's Correlation Coefficient value (PCC <0.3) but found a significant increase in the value (PCC >0.5) in MMS-treated cells (*Figure 4D–F*). We verified the accuracy of the assay by rotating the red and green fluorescence images through 180° relative to each other and recalculating the PCC value as described earlier (*Ma et al., 2023*). After rotation, we failed to observe any significant correlation between Scm3-13Myc and γ-H2A (*Figure 4—figure supplement 1G–I*), suggesting the observed high PCC values (*Figure 4D–F*) are not an artifact of the assay. In order to negate the possibility that the increased co-localization of Scm3-6HA with Rad52 or γ-H2A following MMS treatment is merely due to the increased accumulation of the DDR proteins in response to MMS within the small yeast nucleus, we tested the colocalization of an unrelated transcription factor, Hsf1, with γ-H2A. Since Hsf1 is known to increase its chromatin association in response to heat shock (*Chowdhary et al., 2019*; *Rubio et al., 2024*), we verified this by incubating the cells at 37 °C for 15 min. As expected, Hsf1 showed an increased chromatin association (a greater number of Hsf1-13Myc foci) at high temperatures compared to normal temperatures (*Figure 4—figure supplement 2A–C*). As expected, the increased chromatin occupancy of Hsf1 or γ-H2A due to high temperature or

MMS treatment, respectively, did not result in any significant increase in the co-localization frequency between the two proteins (*Figure 4—figure supplement 2D–F*). Taken together, we conclude that the association of Scm3 with chromatin and its co-localization with DDR proteins increases with DNA damage.

Since we observed high co-localization between Scm3-13Myc and γ-H2A in MMS-treated cells (*Figure 4D*), we wished to understand if the localization of Scm3 is dependent on γ-H2A. For this, we performed chromatin spread assay to visualize the behavior of Scm3 in the *mec1Δ tel1Δ* double mutant, where γ-H2A remains absent on the chromatin at the DSB sites. As expected, we observed a drastic drop in γ-H2A localization on the spreads in the double mutant as compared to wild-type cells in MMS--treated samples, which validates the assay. Upon quantification of the total intensity of Scm3 and normalizing that with the background intensity, we did not observe any statistically significant drop in the localization of Scm3 on the chromatin (*Figure 4—figure supplement 1J–L*). This suggests that the recruitment of Scm3 at the damage sites is not influenced by prior localization of γ-H2A at those sites.

## Scm3 is recruited to a site-specific DSB

The overall increased association of Scm3 with chromatin and its colocalization with Rad52 and γ-H2A in response to MMS-mediated DNA damage prompted us to examine if Scm3 is recruited specifically to DNA damage sites. For this, we utilized the chromatin immunoprecipitation (ChIP) assay to visualize the dynamics of Scm3 recruitment at an induced DSB site. We used a yeast strain (NA14) harboring an HO cut site integrated at the mutant *ura3* locus, which is cleavable by HO endonuclease upon galactose induction (*Agmon et al., 2009*; *Fangaria et al., 2022*). The strain also contains a distant *URA3* locus, which is utilized to repair the cut *ura3* locus post-DSB induction. The cleavage by HO endonuclease was first verified by PCR-based analysis using primers complementary to the DNA sequence flanking the HO cut site (*Figure 5A*). The DNA damage was maximal at 2 hr of galactose induction, and the damage was majorly repaired by 4 hr of galactose induction (*Figure 5—figure supplement 1A–B*).

We monitored the recruitment of Scm3 near DSB and at sites distal to that up to –3 kb away (*Figure 5A*) by performing ChIP of Scm3-6HA from the samples collected at every hr up to 4 hr following galactose induction. As a positive control for Scm3 ChIP, we found significant enrichment of Scm3 at the centromere (*CEN3*; *Figure 5—figure supplement 1C*). We observed a significant increase in the Scm3 binding near the DSB at 2 hr of galactose induction and a decrease in the enrichment at 4 hr, possibly due to the gradual repair of the damage (*Figure 5B–C*). We also observed an increase in the enrichment of Scm3 up to –3 kb distal to the cut site, which decreased gradually over time (*Figure 5C*) and no significant enrichment at the *TUB2* locus, used as negative control (*Figure 5C*) or in the no antibody control (*Figure 5D*). This suggests that Scm3 is maximally recruited to the cleavage sites at the time of DNA damage and gradually dissociates when the DNA is repaired. To confirm that the enrichment of Scm3 at 2 hr at the HO-induced DSB sites is a site-specific response rather than a global chromatin enrichment, we verified the enrichment of Scm3-6HA at the *CEN3* and *TUB2* loci at different time points of HO induction. We could not observe any significant enrichment of Scm3 at those loci over different time points, suggesting Scm3 specifically enriches at the DSB sites upon DNA damage (*Figure 5—figure supplement 1D, E*). To verify that the HO-induced DSB can recruit DNA repair proteins, we also investigated the enrichment of one such protein, Rad51, which is known to bind DSB sites (*Fangaria et al., 2022*). We found Rad51 maximally enriched at the DSB site 2–3 hr after galactose induction (*Figure 5—figure supplement 1F*). We conclude that upon DNA damage, Scm3 associates with the damage sites and may act in cis to facilitate damage repair.

We postulate that the association of Scm3 at the DSB site is to facilitate DNA damage repair. In that case, we expect to observe an altered kinetics of repair in the absence of Scm3. To examine this, we constructed the *SCM3-AID* allele in the NA14 background and released the cells in galactose (for HO induction and DNA cleavage) after treating them with DMSO (wild type) or auxin (Scm3-depleted). Cells were harvested at different time points, as mentioned earlier, and the extent of DNA damage and DNA repair was monitored using primers flanking the HO cut site (*Figure 5A*). While in the wild type, the DSB is repaired by 55% by 4 hr and is completed by 6 hr, the Scm3-depleted cells showed a delay in the DSB repair, which was only 15% by 4 hr time point and reaches a maximum of 65% by 6 hr time point (*Figure 5—figure supplement 2A, B*). This delay in repair is independent of the cell

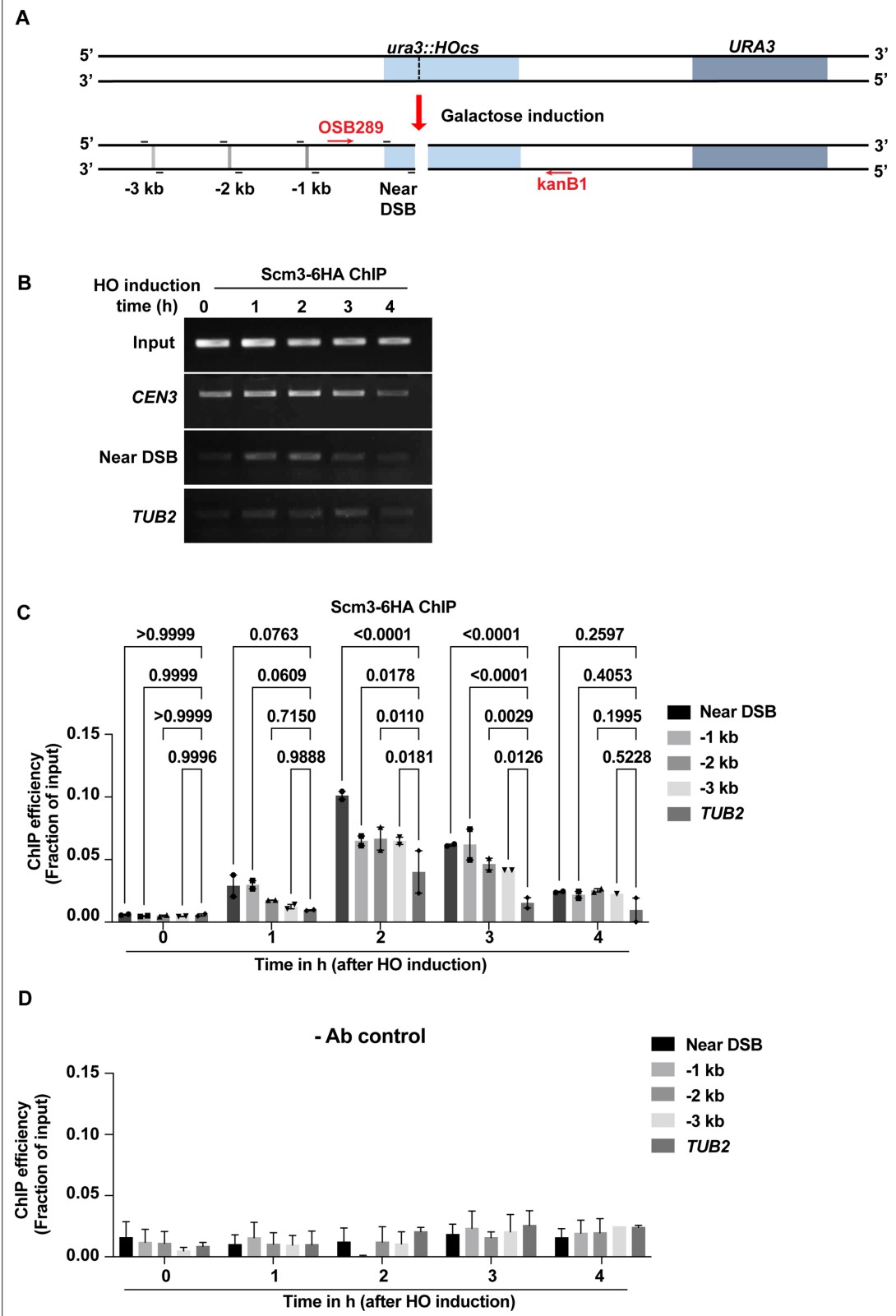

**Figure 5.** Scm3 is recruited to a site-specific double-stranded break (DSB). (**A**) Schematic representation of the *HO* endonuclease cut site (*HOcs*) integrated at the *URA3* locus in the NA14 strain. Upon galactose induction, the *HO* endonuclease cleaves the DNA at the specific site, generating a DSB. The positions of the primers (OSB289 and kanB1), used to detect the DSB, are shown along with primers to detect binding of Scm3 near and distal (−1 to −3 kb) to the DSB (**B**) ChIP analyses were performed to quantify the association of Scm3-6HA in the NA14 strain. The PCR-amplified DNA bands

*Figure 5 continued on next page*

*Figure 5 continued*

following ChIP were run on an agarose gel and stained with ethidium bromide. (**C**) ChIP analyses for measuring the association of Scm3-6HA with the indicated loci were performed at the indicated time points of HO induction. The ChIP efficiency as percentage enrichment per input obtained by qPCR analysis of the immunoprecipitated DNA is plotted. (**D**) The same analysis as in (**C**) using no antibody (-Ab) control. Error bars represent the standard error of the mean. The p-values in (**C**) and (**D**) were determined by two-way ANOVA test followed by Dunnett's multiple comparison test from two independent experiments.

The online version of this article includes the following source data and figure supplement(s) for figure 5:

**Source data 1.** Original files for gels displayed in *Figure 5B*.

**Source data 2.** Original files for gels displayed in *Figure 5B* indicating relevant bands.

**Source data 3.** Source data for *Figure 5*.

**Figure supplement 1.** Dynamics of *HO*-induced DSB formation and protein binding at centromere and DSB sites.

**Figure supplement 1—source data 1.** Original files for gels displayed in *Figure 5—figure supplement 1A*.

**Figure supplement 1—source data 2.** Original files for gels displayed in *Figure 5—figure supplement 1A* indicating relevant bands.

**Figure supplement 1—source data 3.** Source data for *Figure 5—figure supplement 1*.

**Figure supplement 2.** DSB repair is delayed in the absence of Scm3.

**Figure supplement 2—source data 1.** Original files for gels displayed in *Figure 5—figure supplement 2A*.

**Figure supplement 2—source data 2.** Original files for gels displayed in *Figure 5—figure supplement 2A* indicating relevant bands.

**Figure supplement 2—source data 3.** Source data for *Figure 5—figure supplement 2*.

cycle delay mediated by Scm3 depletion, as cells arrested at metaphase also show significant repair in G2/M stage of the cell cycle (*Aylon et al., 2004*). This result suggests that Scm3 at the DSB site aids in the repair of the DNA damage, perhaps by interacting with the DDR proteins (*Figure 4*).

## Activation of the DDR checkpoint is perturbed in the *scm3* mutant

Having shown the recruitment of Scm3 at the DNA damage sites and the observation that Scm3-depleted cells are compromised in repairing DNA damage, we wished to understand if Scm3 being at the damage sites activates the DDR checkpoint, which is required for damage repair. In the DDR pathway, Rad53 is an effector kinase that becomes phosphorylated in a Mec1 sensor kinase-dependent manner in response to DNA damage (*Ciccia and Elledge, 2010*; *Melo and Toczyski, 2002*; *Zhou and Elledge, 2000*). The phosphorylation of Rad53 is essential to halt the cell cycle and provide cells with time to repair damaged DNA. To examine whether Scm3-depleted cells have defects in the activation of DDR checkpoint and in halting the cell cycle, we measured Rad53 phosphorylation in *SCM3-AID* cells in the presence or absence of auxin after treating them with 0.02% MMS for 90 min (*Figure 6—figure supplement 1A*). Anti-Rad53 antibodies that recognize both phosphorylated and unphosphorylated forms of Rad53 were used. We observed lesser phosphorylation of Rad53 in Scm3-depleted cells (*SCM3-AID*+auxin) as compared to *SCM3-AID* cells (*Figure 6—figure supplement 1B–C*).

Since Scm3-depleted cells, unlike the wild type, arrest at the G2/M stage due to activation of spindle assembly checkpoint (SAC) (*Camahort et al., 2007*; *Stoler et al., 2007*), the observed difference in Rad53 phosphorylation could be due to the difference in cell cycle stages. To circumvent this issue, we grew *CDC20-AID* and *SCM3-AID* cells for 3 hr before treating them with 0.75 mM auxin for 2 hr to arrest them at G2/M by depleting Cdc20 and Scm3, respectively (*Figure 6A*). The cells were then treated with 0.02% MMS and harvested at indicated time points for total protein extraction. The *CDC20-AID* and *SCM3-AID* cultures showed around ~80% metaphase arrest cells after 2 hr of auxin treatment (0 min in *Figure 6—figure supplement 1D*). Notably, while the Cdc20-depleted cells maintained the arrest even with a longer incubation in auxin, the Scm3-depleted cells tended to bypass the arrest after a certain time and showed accumulation of multibudded cells with fragmented DAPI, an arrest-release phenotype (*Figure 6—figure supplement 1E*, *Figure 6C*). Consistent with our previous results (*Figure 6—figure supplement 1B–C*), we observed that in *SCM3-AID*+auxin cells, Rad53 was less phosphorylated than in *CDC20-AID*+auxin cells (*Figure 6B*). Since Scm3-depleted cells showed reduced Rad53 phosphorylation, we wished to examine if the lack of Scm3 perturbs Mec1 kinase function, which phosphorylates Rad53. Mec1 also phosphorylates histone H2A at S129 upon DNA damage (*Lee et al., 2014*). However, we failed to observe any significant difference in the phosphorylation status of H2AS129 between Scm3- or Cdc20-depleted cells (*Figure 6—figure supplement 1F*),

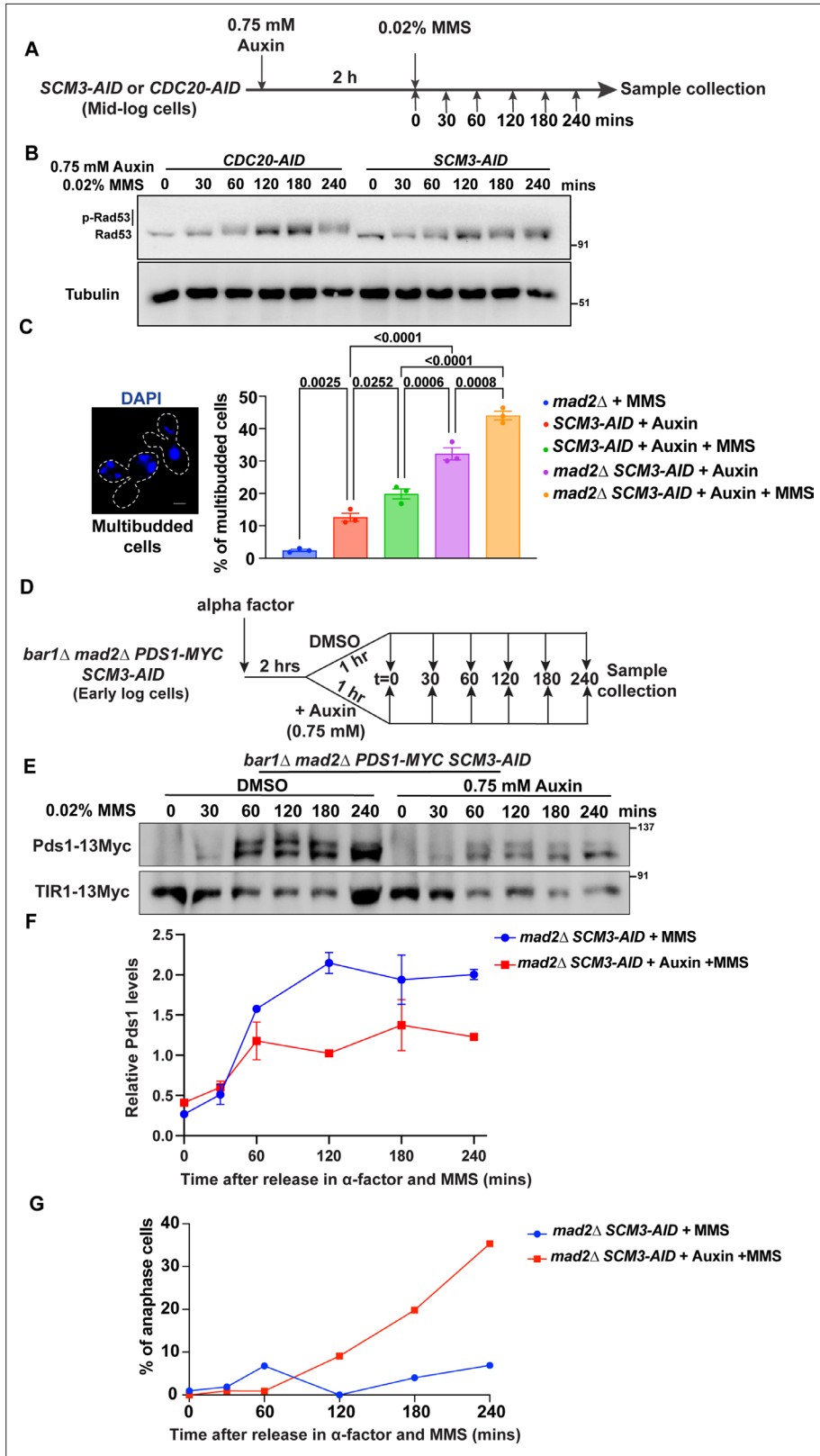

**Figure 6.** Evidence of a DNA damage checkpoint defect in the absence of Scm3. (**A**) Schematic of the experimental strategy to detect Rad53 in Scm3- or Cdc20-depleted cells treated with MMS for different time points. (**B**) Western blots from indicated cells according to the protocol in (**A**). (**C**) *mad2Δ, SCM3-AID,* and *SCM3-AID mad2Δ* cells were first treated with 0.75 mM auxin for 2 hr to deplete Scm3 and then either treated with 0.02%

*Figure 6 continued on next page*

*Figure 6 continued*

MMS for 90 min or left untreated. The cells were harvested and scored for the presence of multibudded cells (shown on the left). At least 150 cells from three independent experiments were analyzed for each set. Error bars represent the standard error of the mean. The p-values were determined by a one-way ANOVA test followed by Tukey's multiple comparison test. Scale bar = 2 μm. (**D**) Schematic of the experimental strategy to detect Pds1 in *mad2Δ SCM3-AID* cells in the presence of DMSO or auxin (Scm3-depleted) and each treated with 0.02% MMS for different time points. (**E**) Western blot showing Pds1 and TIR1 (as loading control). Cells were harvested following the strategy given in (**D**). (**F**) The intensities of the Pds1 measured by ImageJ were plotted after normalizing with the corresponding loading control (TIR1). (**G**) The percentage of anaphase cells judged by DAPI staining and bud morphology for each time point of cells harvested in (**D**) is presented graphically.

The online version of this article includes the following source data and figure supplement(s) for figure 6:

**Source data 1.** Original files for western blot analysis displayed in *Figure 6B and E*.

**Source data 2.** Original files for western blot analysis displayed in *Figure 6B and E* indicating relevant bands.

**Source data 3.** Source data for *Figure 6*.

**Figure supplement 1.** H2A phosphorylation is not perturbed in the absence of Scm3.

**Figure supplement 1—source data 1.** Original files for western blot analysis displayed in *Figure 6—figure supplement 1B, F*.

**Figure supplement 1—source data 2.** Original files for western blot analysis displayed in *Figure 6—figure supplement 1B, F* indicating relevant bands.

**Figure supplement 1—source data 3.** Source data for *Figure 6—figure supplement 1*.

indicating that Mec1 function is not perturbed in the absence of Scm3. These results indicate that in the absence of Scm3, when there is DNA damage, the activation of the DDR checkpoint is improper, and consequently, the phosphorylation of Rad53 is compromised.

If Scm3-depleted cells fail to activate the DNA damage checkpoint, we would expect a failure to arrest the cell cycle upon DNA damage, resulting in multibudded cells. However, Scm3-depleted cells arrest at G2/M due to improper kinetochore-microtubule attachment which activates the spindle assembly checkpoint (SAC; *Camahort et al., 2007*; *Stoler et al., 2007*). We therefore deleted the SAC component *MAD2* in *SCM3-AID* cells to exclude the effect of a SAC-mediated arrest. These cells were first depleted of Scm3 by treatment with auxin and then, along with *mad2Δ* control cells, incubated with or without MMS for 90 min (*Figure 6C*). While the *mad2Δ* cells in the presence of MMS showed few multibudded cells (<5%), the percentage of multibudded cells strongly increased after depletion of Scm3 (>40%), suggesting that Scm3-depleted cells failed to activate the DNA damage checkpoint in the presence of DNA damage. To further validate this result, we monitored the degradation kinetics of Pds1 in the background of *MAD2* deletion. *mad2Δ SCM3-AID* cells were first arrested in G1 by alpha factor and then released in MMS-containing media in the absence or presence of auxin (*Figure 6D*). Samples were harvested at regular intervals, and Pds1 levels were monitored by western blotting. We could observe lower Pds1 levels and its early removal in the *mad2Δ SCM3-AID* (+auxin) cells compared to *mad2Δ SCM3* (DMSO) cells (*Figure 6E and F*). This was further verified by an early appearance of anaphase cells (large-budded cells with DAPI in the bud) in the Scm3-depleted cell population (*Figure 6G*), indicating that these cells are not able to effectively arrest the cell cycle at G2/M in response to DNA damage.

## Phosphorylation of Scm3 in response to DNA damage

The function of Scm3 in DDR could be mediated by a damage-induced post-translational modification of this protein. To examine if Scm3 becomes modified upon DNA damage, we treated *SCM3-6HA* cells with increasing concentrations of MMS or with 0.02% MMS for increasing duration. We observed slower migrating bands of Scm3 (Scm3*) that became brighter with increasing concentrations of MMS and were present very faintly, even in the untreated cells (*Figure 7A*, left panel). We observed an increase in the background-subtracted[2]intensity of Scm3* and Scm3*/Scm3 ratio with increasing MMS concentration (*Figure 7A*, middle, and right panels). A similar observation was made when the cells were treated with 0.02% MMS for increasing time (*Figure 7—figure supplement 1A*). Since we could also detect Scm3* bands in the untreated cells, its enrichment in response to MMS could be due to a cell cycle arrest caused by MMS. To understand the dynamics of the appearance of Scm3*

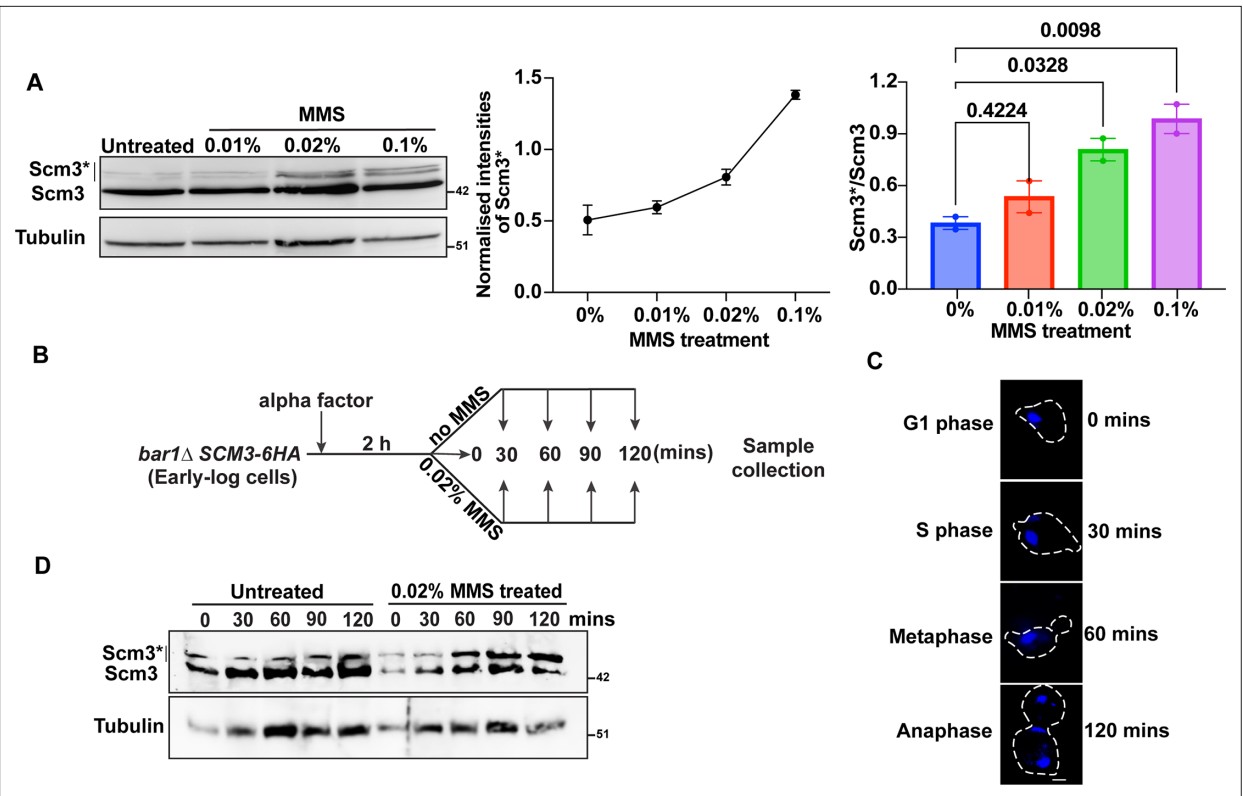

**Figure 7.** Post-translational modification of Scm3 in response to DNA damage. (**A**) Left, western blot showing Scm3-6HA and its modified form (Scm3*) along with tubulin as loading control from the cells untreated or treated with indicated concentrations of MMS for 90 min. Middle, intensity of Scm3* with increasing MMS concentrations, normalized to tubulin. Right, the ratio of the modified to unmodified forms of Scm3. Error bars represent the standard error of the mean obtained from two independent experiments. The p-values were determined by a one-way ANOVA test followed by Dunnett's multiple comparison test. (**B**) Schematic of the experimental strategy that was followed for detecting Scm3-6HA in G1 arrested cells and following their release from the arrest in the absence or presence of MMS. (**C**) Representative images showing the cell cycle stages by DAPI staining and bud morphology at indicated time points after release from G1 arrest. (**D**) Western blot showing Scm3-6HA along with tubulin as a loading control from the cells harvested, as shown in (**B**). The cell lysates were probed using anti-HA and anti-tubulin antibodies to detect Scm3-6HA and tubulin, respectively. Scale bar = 2 μm.

The online version of this article includes the following source data and figure supplement(s) for figure 7:

**Source data 1.** Original files for western blot analysis displayed in *Figure 7A and D*.

**Source data 2.** Original files for western blot analysis displayed in *Figure 7A and D* indicating relevant bands.

**Source data 3.** Source data for *Figure 7*.

**Figure supplement 1.** Phosphorylation of Scm3 in response to DNA damage.

**Figure supplement 1—source data 1.** Original files for western blot analysis displayed in *Figure 7—figure supplement 1A, D, E, F, G*.

**Figure supplement 1—source data 2.** Original files for western blot analysis displayed in *Figure 7—figure supplement 1A, D, E, F, G* indicating relevant bands.

**Figure supplement 1—source data 3.** Source data for *Figure 7—figure supplement 1*.

bands over different cell cycle stages and in response to MMS, we released the *bar1Δ SCM3-6HA* cells from alpha-factor-mediated G1 arrest into medium in the absence or presence of MMS (*Figure 7B*). We harvested the cells at indicated time points and evaluated the cell cycle synchrony using bud morphology and DAPI staining (*Figure 7C*, *Figure 7—figure supplement 1B-C*). The majority of cells at 0, 30, 60, 90, and 120 min are in G1, S, metaphase, metaphase/anaphase, and anaphase, respectively. We could faintly observe the Scm3* bands after release from the G1 phase in MMS-free (untreated) media, and the band intensity did not change over time. However, when the cells were released in the presence of MMS, we observed an increase in the intensity of the Scm3* bands over time (*Figure 7D*). These results suggest that the accumulation of Scm3* in the MMS-treated cells is due to DNA damage rather than an effect of the pace of the cell cycle. It is possible that Scm3* bands

are phosphorylated forms of Scm3 since the protein has several serine/threonine residues. To test this, we challenged the Scm3-6HA cells with MMS, as in *Figure 7A*, and resolved the proteins on SDS-PAGE with or without Phos-Tag reagent. The presence of Phos-Tag in the gel enables slow mobility of the phosphorylated proteins and thus further differentiates them from their unphosphorylated forms. The mobility of the Scm3* bands was further reduced with respect to the Scm3 band in Phos-Tag gel compared to the normal gel (*Figure 7—figure supplement 1D–E*), suggesting that the modification is likely due to phosphorylation. We further verified this by performing a phosphatase experiment. *SCM3-6HA* cells treated or untreated with MMS were immunoprecipitated and subjected to alkaline phosphatase treatment for 3 hr at 37 °C. The disappearance of the Scm3* band upon phosphatase treatment confirms that the shift was indeed due to phosphorylation (*Figure 7—figure supplement 1F*). Notably, in several high-throughput studies, Scm3 has been shown to be phosphorylated at the S39, S64, and S165 residues (*Lanz et al., 2021*; *MacGilvray et al., 2020*; *Zhou et al., 2021*). The S64 phosphorylation of Scm3 was found to be enriched in cells after MMS treatment for 2 hr as compared to G1 arrested cells, although the authors could find S64 phosphorylation in only two out of four replicates used in their study (*Lanz et al., 2021*). However, in the absence of further validation in those studies, our observation of a gradual increase in the Scm3* band intensity with increasing dose of MMS convincingly demonstrates DNA damage-dependent phosphorylation of Scm3.

Since Mec1 and Tel1 kinases are the sensor kinases that contribute to the activation of the DDR pathway through protein phosphorylation, we tested if Scm3 undergoes a Mec1- and/or Tel1-dependent phosphorylation in response to MMS. We tested the phosphorylation status of Scm3-6HA in the single (*sml1Δ mec1Δ, sml1Δ tel1Δ*) and also in the double (*sml1Δ mec1Δ tel1Δ*) kinase mutants in the presence of MMS. We could not observe any loss of Scm3 phosphorylation in either of the

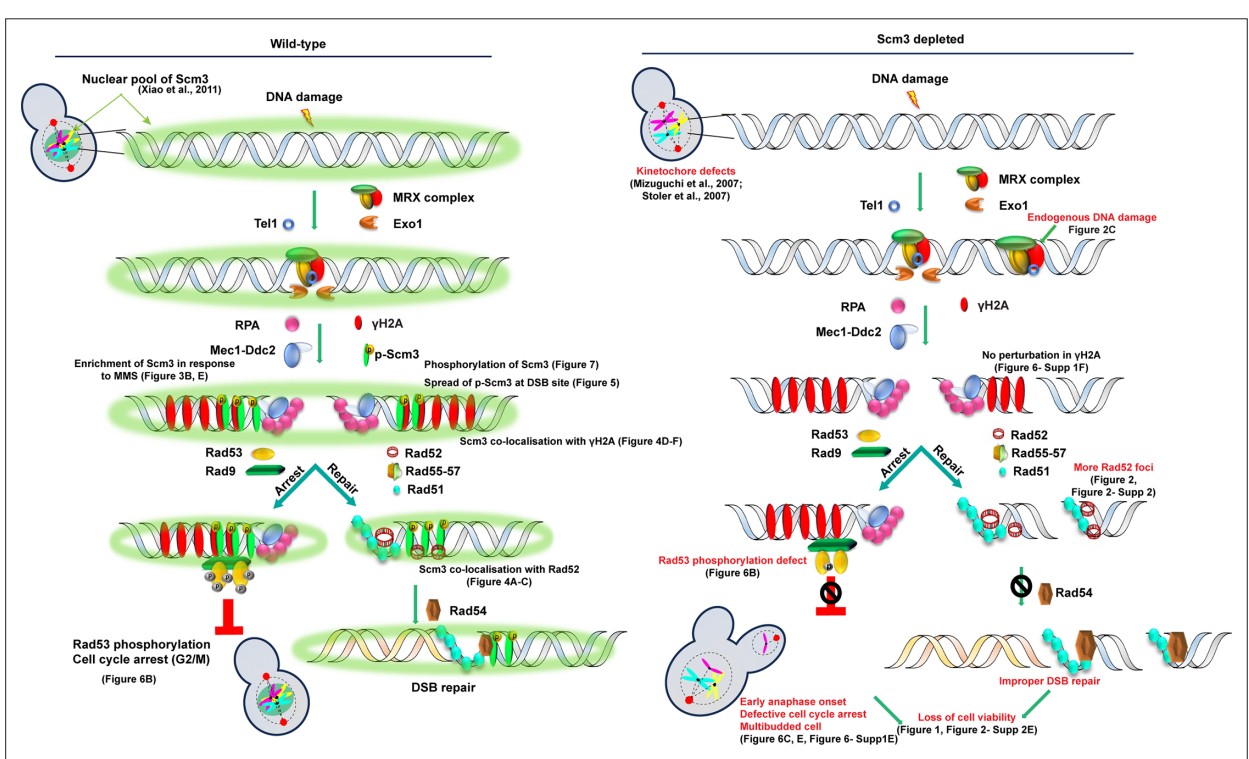

**Figure 8.** A working model summarizing the role of Scm3 in the DNA damage response (DDR). Left, in wild-type cells, a subset of Scm3 present in the nucleus (green) is phosphorylated upon DNA damage. The phosphorylated Scm3 (p-Scm3) associates with DSB sites, colocalizes with γ-H2A, and spreads to chromatin flanking the DSB site. Possibly, p-Scm3 facilitates proper activation of Rad53 through its phosphorylation, leading to cell cycle arrest. Additionally, p-Scm3 co-localizes with Rad52 and assists in DSB repair. Right, in Scm3-depleted cells, there is an increase in endogenous DNA damage, corresponding to more Rad52 foci, likely due to the inability to repair DSBs, and cell viability drops. In parallel, reduced Rad53 activation causes improper DDR-mediated cell cycle arrest, generating multibudded cells and possibly aneuploid cells, leading to cell death. Phenotypes associated with the loss of Scm3 are shown in red. Graphical elements in the figure are partially adapted from the Servier Medical Art Repository (https://smart.servier.com), licensed under CC BY 4.0 (https://creativecommons.org/licenses/by/4.0/).

mutants (*Figure 7—figure supplement 1G*), suggesting at least the sensor kinases Mec1 or Tel1 are not responsible for Scm3 phosphorylation.

## Discussion

The bona fide function of Scm3 is to deposit Cse4 at the centromeres during S phase in budding yeast to facilitate kinetochore formation. The observation that Scm3 persists not only at the centromeres throughout the cell cycle but also associates with the bulk of the chromatin argues for additional kinetochore-independent functions of Scm3 in maintaining genome stability. In metazoans, the Scm3 homolog HJURP was initially identified as a protein involved in DDR, and its role in CENP-A deposition was only discovered later. How HJURP regulates the DDR pathway is not well understood. The DDR role of Scm3 has not previously been reported in budding yeast to our knowledge. Showing that Scm3 also contributes to the DDR pathway in budding yeast is important since this organism is distinct from metazoans in terms of harboring small 125 bp point centromeres, contrasting the megabase pair large regional centromeres in metazoans, and its centromeres are not epigenetically determined. Here, for the first time, we report that Scm3, perhaps upon phosphorylation, promotes DNA damage repair by contributing to the activation of the DNA damage checkpoint, and in the process, it possibly associates with DDR proteins at the DNA damage sites (*Figure 8*). Our findings significantly advance knowledge of how CENP-A chaperones can be involved in the DDR pathway.

### Lack of Scm3 increases DNA damage sensitivity

It is technically challenging to perform a growth-based sensitivity assay using cells depleted for an essential gene. To circumvent this issue with Scm3, which is essential for cell survival, we partially depleted Scm3 and found that these cells were sensitive to genotoxic agents, including MMS, HU, and CPT (*Figure 1*, *Figure 1—figure supplement 2A*). The observed sensitivity is not due to problems in kinetochore formation or cell cycle progression in the absence of Scm3, as other kinetochore mutants or a metaphase block in wild-type cells did not show similar sensitivity (*Figure 1—figure supplement 3*). The arguments favoring the possible centromere-independent DDR functions of Scm3 can be gleaned as follows. From comparing the domain organization of Scm3 with HJURP (*Figure 1—figure supplement 1A*), we noticed that, besides the CENP-A/Cse4 binding domain, both HJURP and Scm3 have a common DNA binding domain, which is dispensable for CENP-A/Cse4 binding and also dispensable for Mis18 or Ndc10 binding in metazoans or budding yeast, respectively. Both in vitro and in vivo evidence demonstrate binding of the DNA binding domain to DNA (*Kato et al., 2007*; *Mizuguchi et al., 2007*), suggesting this domain can recruit Scm3 to non-centromeric DNA damage sites. In support of this, Scm3 has recently been shown to be phosphorylated in an MMS-dependent manner at S64 (*Lanz et al., 2021*), which, interestingly, lies within the DNA binding domain. Among other non-essential regions of Scm3, neither the NLS region nor the C-terminal region is involved in DDR (*Figure 1—figure supplement 1B*). On the other hand, the D/E-rich region, which is important for Scm3 stability (*Stoler et al., 2007*), forms a disordered domain as per alpha-fold prediction and hence is believed to be capable of interacting with multiple proteins or can act as a scaffold for engaging several protein-protein interactions which are required in DDR. It will be interesting to test whether and how CENP-A might influence the DDR function of Scm3. Analysis of damage-induced localization of CENP-A and isolation of separation-of-function (centromeric vs non-centromeric) mutants of Scm3 could provide insights into the role of CENP-A in Scm3's DDR function.

MMS halts replication forks due to the formation of DNA adducts, which eventually generate recombinogenic lesions (*Conde and San-Segundo, 2008*). In yeast, these lesions are repaired primarily through homologous recombination (HR), facilitated by the Rad52 epistasis group of proteins (*Pizzul et al., 2022*; *Sirbu and Cortez, 2013*). In the presence of DNA damage, Rad52 relocalizes to form bright foci, which mark the repair centers. Strikingly, we observed that the number of Rad52 foci is increased in the absence of Scm3, even without external DNA damage (*Figure 2C*, Left), with maximal accumulation noticed at 60 min post-G1 release, which coincides with the timing of DNA replication (*Figure 2—figure supplement 2C*). This indicates that Scm3 might be required to avoid generating DNA lesions during replication, which itself is genotoxic. In support of this, it was previously observed that cells arrest during S phase in the absence of Scm3 (*Camahort et al., 2007*). In yeast, although HR is the preferred repair pathway, an alternative pathway named non-homologous end joining (NHEJ)

exists. The absence of epistatic interaction between *SCM3* and *RAD52* (*Figure 1C*) suggests that Scm3 may function in ways other than the Rad52-mediated classical HR pathway. In this context, it would be interesting to test how Scm3 might interact with the key proteins of the NHEJ pathway, such as Ku70/Ku80 and Lig4 (*Gao et al., 2016*). It is possible that Scm3 may promote a specific chromatin architecture facilitating that DSB ends stay together and are accessible for NHEJ-mediated end joining.

## Association of Scm3 with non-centromeric loci

In budding yeast, chromosome XII harbors ribosomal DNA (rDNA) with around 150 repetitive loci (*Dammann et al., 1993*; *Smith, 2022*). Recombination among these repeats excises rDNA circles, and therefore, efficient DNA damage repair through HR at these loci is crucial for preserving genomic stability. Notably, mammalian HJURP binds to the rDNA region and maintains its stability (*Kato et al., 2007*). The knockdown of HJURP using siRNA resulted in increased rDNA recombination events and chromosome instability (*Kato et al., 2007*). The presence of Scm3 at the rDNA loops (*Figure 3A and F*) suggests that Scm3, like HJURP, might also be important in regulating the recombination and repair processes at these rDNA loci.

During DSB repair, lesions generated at multiple sites are recruited to repair centers, which are marked by Rad51 and Rad52 (*Lee et al., 2003*; *Lisby et al., 2001*; *Miyazaki et al., 2004*). The observed statistically significant co-localization of Rad52 and γ-H2A with Scm3 in response to DNA damage (*Figure 4A*) implies the recruitment of Scm3 at multiple DNA damage sites distributed over several repair centers. Normally, these proteins localize independently onto chromatin owing to their specific functions and show limited overlap (*Figure 4—figure supplement 1*), but in response to DNA damage, they significantly co-localize with each other aiding in the process of damage repair. This was supported by an overall increase in the association of Scm3 with the non-centromeric chromatin sites, the majority of which, through in-silico analysis, appear to be DNA-damage-prone sites (*Figure 3—figure supplement 1*). At the damage sites, it is possible that Scm3 might physically interact with Rad51/Rad52 and/or MRX (Mre11, Rad50, Xrs2) complex through its HR or DNA binding or the D/E rich domain (*Figure 1—figure supplement 1*). However, we failed to observe any interaction of Scm3 at least with Rad52 using a co-immunoprecipitation.

Although DNA damage sites are marked by γ-H2A in budding yeast, γ-H2A is also observed at stalled replication forks, fragile sites, telomeric, and heterochromatin regions (*Kirkland et al., 2015*; *Kitada et al., 2011*; *Bewersdorf et al., 2006*). Therefore, the increased co-localization between Scm3 and γ-H2A upon MMS treatment (*Figure 4D–F*) suggests that Scm3 may be recruited to those sites as well. The presence of Scm3 up to 3 kb distal from the cut site (*Figure 5C*) is consistent with the pattern for γ-H2A, which also spreads distal to the cut site (*Lee et al., 2014*). Though Scm3's presence at the cut site peaked at 2 hr following HO induction, it persisted at the cut site even when ~60% of the cut DNA was repaired, indicating that Scm3 remains associated with damage sites during the repair process. Importantly, in the absence of Scm3, the repair of the cut site was drastically reduced, suggesting that the localization of Scm3 at the cut site is required for the repair of the damage. These findings suggest that the localization of Scm3 at the DSB sites and its spreading up to –3 kb promote the repair process. Although how Scm3 might promote repair is currently not understood, it may facilitate multiple steps in the repair process, which will be investigated in future studies.

## Scm3 regulates the DNA damage checkpoint

We wanted to address the mechanism by which Scm3 might be involved in DDR. Inefficient Rad53 phosphorylation (*Figure 6B*) and abrogation of the cell cycle arrest (*Figure 6E and C*) in the presence of DNA damage in Scm3-depleted cells suggest that these cells are defective in DDR checkpoint activation. Since Rad53 is known to be phosphorylated in a Mec1-dependent manner (*Gilbert et al., 2001*; *Pellicioli and Foiani, 2005*), we examined but failed to observe any attenuation of Mec1 activity in Scm3-depleted cells in the presence of DNA damage (*Figure 6—figure supplement 1F*). Therefore, we are uncertain about the cause of improper Rad53 phosphorylation; a defect in its auto-activation or in the activity of other mediator kinases, for example Tel1, may be responsible. Nevertheless, given the increased phosphorylation of Scm3 in response to DNA damage, phosphorylated Scm3 may be required to activate Rad53 (*Figure 7A*). Although we found that Scm3 phosphorylation is Mec1 independent (*Figure 7—figure supplement 1G*), other kinases may be involved. While the involvement

of CDKs in the DNA damage checkpoint is not well understood, it is known that CDKs contribute to the phosphorylation of both Rad52 and Rad51 to promote repair (*Lim et al., 2020*). Given that the centromeric association of Scm3 varies with the cell cycle stage (*Mishra et al., 2011*), probably due to oscillating levels of CDK activity, the DNA damage-dependent phosphorylation of Scm3 might also be regulated by CDKs. However, this does not rule out the involvement of other kinases, including Rad53 and Tel1, which are closely associated with the DNA damage checkpoint pathway. Further investigation is required to determine the kinases involved in the Scm3 phosphorylation.

In summary, we report here that, like HJURP, Scm3 in budding yeast also possesses functions in DDR, indicating that the centromere-independent functions of the CENP-A chaperone are evolutionarily conserved. Although the chromatin landscape of budding yeast is far more 'open' with lesser involvement of epigenetic determinants than in metazoans, the inherent similarity in the DNA damage repair pathway may necessitate the utilization of these chaperones in a similar fashion in both cell types. Not much is known about how HJURP is involved in the DDR pathway. As we find that Scm3 is involved in the proper activation of the DNA damage checkpoint (*Figure 8*), this could be tested for HJURP in metazoans. Abrogation of this surveillance function may lead to aneuploidy and disease states. Our findings on Scm3 will pave the way for future work using yeast as a model to decipher genome-wide functions of the CENP-A chaperone, like HJURP, which is crucial given its genome-wide localization and evidence of its modulation in several patients.

## Materials and methods

### Yeast strains

All *Saccharomyces cerevisiae* strains used in this study are derived from W303 genetic background and are mentioned in *Supplementary file 1A*. C-terminal tagging (AID, HA, Myc, GFP) or gene deletions were performed using PCR-based integration at endogenous locations as described before (*Janke et al., 2004*; *Longtine et al., 1998*; *Wach, 1996*). Standard lithium acetate-based yeast transformation methods were employed for all genetic manipulations (*Daniel Gietz and Woods, 2002*).

### Media, reagents, and growth conditions

Yeast strains were grown using yeast extract, peptone, and dextrose (YPD; 1% wt/vol yeast extract, 2% wt/vol peptone, 2% wt/vol dextrose for liquid and an additional 2% % wt/vol agar for plates) at 30 °C (*Sherman, 2002*). For plates containing MMS, HU, and CPT (Sigma Chemicals, Switzerland), the drugs were added to YPD plates at appropriate concentrations. To arrest the cells at the G1 or metaphase stage of the cell cycle, alpha factor (T6901, Sigma Chemicals, Switzerland) or nocodazole (M1404, Sigma Chemicals, Switzerland) was added to the cells at the indicated concentrations. For HO endonuclease induction, strains were grown in 1% yeast extract, 2% peptone, containing 3% glycerol before adding 3% galactose. For western blotting, immunoprecipitation, or immunofluorescence studies, the primary antibodies used were rat anti-HA (3F10, Roche), mouse anti-HA (12CA5, Roche), mouse anti-Myc (9E10, Roche), rat anti-tubulin (MCA78G, Serotec), rabbit anti-Rad53 (ab104232, Abcam), rabbit anti-Rad51 (PA5-34905, Invitrogen), and rabbit anti-Histone H2A (phospho-S129; ab15083, Abcam). The secondary antibodies used were Rhodamine (TRITC)-conjugated goat anti-Rat (1:200, Jackson), Alexa Fluor 488-conjugated goat anti-Mouse IgG (1:200, Jackson), Rhodamine (TRITC)-conjugated goat anti-Rabbit (1:200, Jackson), HRP-conjugated goat anti-mouse IgG (1:200, Jackson), HRP-conjugated goat anti-rat (1:200, Jackson), HRP-conjugated goat anti-rabbit (1:200, Jackson).

### Cell spotting and viability assay

The cells were grown in yeast extract, peptone, and dextrose (YPD) at 30 °C conditions till the mid-log phase. The cultures were then 10-fold serially diluted and spotted on the plates containing auxin (dissolved in DMSO) or other DNA-damaging agents. All the reagents were added to YPD or specific dropout plates. After spotting, the plates were incubated at 30 °C for 2–3 days before they were photographed, depending on the growth rate.

For treatment of cells in liquid culture, mid-log grown cells were treated with auxin (dissolved in DMSO) or MMS for the indicated periods before they were harvested and spotted on YPD plates or plates containing DNA damaging agents. To measure cell viability, an equal number of cells were

plated on drug-free or drug-containing YPD plates. The cells were grown at 30 °C for 2 days, and the number of colonies was counted. The colonies formed on the drug-free plate were taken as 100% viability.

## G1 arrest and release

Cells were arrested in G1 phase by adding 0.5 μg/ml α-factor to early log phase cells in liquid culture (*Amon, 2002*). The cells were incubated for 2–3 hr or until at least more than 90% of cells showed shmoo morphology. For the release of the cells from α-factor arrest, the cells were harvested and washed twice with pre-warmed water at 30 °C which was followed by washing twice with pre-warmed α-factor-free media at 30 °C with at least 10 times the volume of the initial culture. The cells were then released in α-factor-free media, and samples were collected at different time points.

## Fluorescence imaging

For live cell imaging, the samples were harvested at indicated time points and washed twice in 0.1 M phosphate buffer before imaging. The cells were processed for DAPI staining, as mentioned elsewhere (*Mittal et al., 2020*; *Shah et al., 2023*). Typically, the cells were fixed with 4% formaldehyde at RT for 5 min. The fixed cells were washed twice with 0.1 M phosphate buffer (pH 7.5), vortexed in freshly prepared 50% ethanol for 30 s, and rewashed with 0.1 M phosphate buffer. Before imaging, the cells were resuspended in freshly prepared DAPI (1 μg/ml, Invitrogen) solution in phosphate buffer for 20 min. The images were acquired with a Zeiss Axio Observer Z1 fluorescence microscope (63×or 100×[3]1.4 NA) in z-stack mode (0.2–0.5 μm spacing).

## Chromatin spreads

Chromatin spreads were performed, as described previously (*Ma et al., 2023*; *Mehta et al., 2014*; *Shah et al., 2023*). Briefly, mid-log (O.D.$_{600}$ = ~1.0) grown cells were harvested and washed once with spheroplasting buffer (1.2 M sorbitol in 0.1 M phosphate buffer). The cells were resuspended in 0.5 ml of the same buffer supplemented with 5 μl of β-mercaptoethanol and 12.5 μl of zymolyase 20T (10 mg/ml, MP Biomedicals, USA) and incubated at 30 °C for 1 h. Once 80–90% of the cells were spheroplasted (by observing under the microscope), the spheroplasting was stopped by adding 1 ml of ice-cold stop solution (0.1 M MES- pH 6.4, 1 mM EDTA, 0.5 mM MgCl$_2$, 1 M Sorbitol). The spheroplasts were centrifuged at 2000 rpm for 2 min and resuspended in 120 μl of ice-cold stop solution. About 60 μl of spheroplasts were mounted on an acid-washed clean glass slide. The cells were treated with freshly made paraformaldehyde solution (4% paraformaldehyde, 3.4% sucrose), followed by 1% Lipsol (LIP Equipment and Services) solution for cell lysis. The lysed cells were homogenously spread over the slide and air-dried at RT. After overnight drying, the slides were treated with 0.4% Kodak Photoflow-200 to avoid photobleaching, followed by 5% skim milk as a blocking solution, which was covered with coverslips. The coverslips were removed, and the slides were incubated with primary and secondary antibodies (1:200) for 60 min with three times PBS washing between antibody treatments. The slides were then incubated with 100 μl of DAPI (1 μg/ml, Invitrogen) for 30 min. After PBS wash for 5 min, 100 μl of mounting solution (90% glycerol supplemented with 1 mg/ml p-phenylenediamine) was added onto the slides. A clean coverslip was placed over a slide and was sealed with transparent nail paint.

## Microscopic image analysis

Images were processed using Zeiss Zen 3.1 software. The z-stacks with the best signal for a particular channel were extracted and merged with the best stack for the other channel. For intensity calculation, a region of interest (ROI) was drawn around the Scm3/Ndc10/γ-H2A+DAPI signal, and the intensity of Scm3/Ndc10/γ-H2A was measured from each chromatin mass or spread (DAPI). An ROI of the same size was put elsewhere in the background area, from where intensities were calculated. The average of the background intensities was then subtracted from the Scm3/Ndc10/γ-H2A intensity obtained from the same spread to get the background-subtracted intensity depicting each dot in the box plot of the respective figures as mentioned previously (*Mittal et al., 2020*; *Shah et al., 2023*).

For Pearson's correlation coefficient (PCC) values, the merged z-stacks and the 'Coloc' tool from the 'Imaris 8.0.2' software were used. The 'automatic thresholding option' in the 'Coloc' tool was used to calculate the threshold for each fluorescence emission channel. The PCC values were recorded

separately for each individual spread. The colocalization was described as no (PCC <0.1), partial (PCC = 0.3–0.5) and complete (PCC >0.5; *Ma et al., 2023*; *Prajapati et al., 2018*; *Zinchuk and Grossenbacher-Zinchuk, 2014*).

## Induction of a site-specific DSB

NA14 (*Fangaria et al., 2022*) cells were grown till 0.3 OD in the presence of 3% glycerol. 5 OD of the cells were harvested (0 hr), and the remaining cells were treated with 3% galactose for 4 hr. 5 OD cells were harvested every hour till 4 hr. The cells were lysed using glass beads in lysis buffer (2% Triton X-100, 1% SDS, 100 mM NaCl, 10 mM Tris pH 8, and 1 mM EDTA pH 8.0), total DNA was isolated as described before (*Hoffman, 1997*) and was solvent extracted using phenol-chloroform-isoamyl alcohol (PCI). The genomic DNA was precipitated and finally resuspended in 30 µl 1 X TE. The concentration of DNA was measured using a nano spectrophotometer. About 30 ng of DNA was used for PCR to detect the DNA break using primers OSB289 and kanB1 (*Fangaria et al., 2022*). Primers against *TUB2* ORF were used as a control.

## ChIP assay and qPCR quantification

ChIP assay was performed as described previously (*Fangaria et al., 2022*; *Kumar et al., 2021*; *Makrantoni et al., 2019*; *Mehta et al., 2014*). First, cells were grown in the presence of 3% glycerol till 0.3 OD. 60 OD cells were harvested (0 hr), and the remaining cells were treated with 3% galactose. 60 OD of cells were harvested every hr till 4 hr. The cells were then fixed with 1% formaldehyde for 30 min at 25 °C at 100 rpm, followed by quenching by adding glycine to a final concentration of 125 mM. The cells were then harvested and washed twice in ice-cold TBS buffer (20 mM Tris–HCl pH 7.5, 150 mM NaCl) and once with 10 ml of FA lysis buffer (50 mM Hepes–KOH pH 7.5, 150 mM NaCl, 1 mM EDTA, 1% Triton X-100, 0.1% Na-deoxycholate) supplemented with 0.1% SDS. The pellet was resuspended in 500 µl of 1 X FA lysis buffer supplemented with 0.5% SDS, 1 mM PMSF, 1 X protease inhibitor cocktail (PIC). The cells were lysed using 0.5-mm glass beads using a mini-bead beater (BIOSPEC, India) for seven cycles (1 min ON, 2 min OFF on ice). The lysate was collected in a pre-chilled 1.5 ml Eppendorf tube and centrifuged twice at 16,000 x *g* for 15 min at 4 °C. The pellet was resuspended in 300 µl ice-cold 1 X FA lysis buffer supplemented with 0.1% SDS, 1 mM PMSF, and 1 X PIC. The chromatin was sheared to 200–600 bps using a water bath sonicator (Diagenode SA, Picoruptor, BC 100, 27 LAUDA Germany) for 30 cycles of 30 s ON / 30 s OFF at 4 °C. After clarification of the lysate by centrifugation at 16,000 x *g* for 10 min at 4 °C, 3–5 µg of appropriate antibodies were added and incubated at 4 °C overnight with gentle rotation. Subsequently, 50 µl Protein-A conjugated Sepharose beads (17-0780-1, GE Healthcare) were added and incubated for 2 hr at 4 °C with rotation. The beads were washed twice with IP wash buffer I (1 X FA lysis buffer, 0.1% SDS, 275 mM NaCl), followed by one wash in IP wash buffer II (1 X FA lysis buffer, 0.1% SDS, 500 mM NaCl, III 10 mM Tris pH 8.0, 250 mM LiCl, 0.5% NP-40, 0.5% sodium deoxycholate, 1 mM EDTA), and 1 X TE at room temperature. The beads were resuspended in elution buffer, and the chromatin was eluted by boiling at 65 °C. The eluate was then decrosslinked overnight at 65 °C, followed by Proteinase K (SRL, India) treatment for 2 hr at 45 °C. The DNA was purified by PCI-based purification and precipitated overnight in chilled ethanol at –80 °C. The enrichment of obtained chromatin fragments was estimated using qPCR (Bio-Rad CFX96, USA) using specific primers targeting specific and non-specific (negative control) chromatin loci, listed in *Supplementary file 1B*. As described before (*Kumar et al., 2021*; *Shah et al., 2023*), the following equation was used to estimate the percentage of chromatin enrichment. ChIP efficiency = Enrichment/Input X 100; Enrichment/Input = $E^{\wedge}\text{-}\Delta CT$; $\Delta CT = CT(ChIP) - [CT(Input) - LogEX(D)]$; E=primer efficiency value, CT = Threshold values obtained from qPCR, D=Input dilution factor. E was estimated as $\{[10^{\wedge}(-1/slope)]- 1\}$ from standardization graphs of CT values against dilutions of the input DNA.

## Chromatin immunoprecipitation (ChIP) and sequencing

ChIP assay was performed as described above with modifications. The cells were grown till mid-log phase and treated with nocodazole (20 µg/ml) for 90 min to arrest cells at metaphase. Cells were subsequently treated with 0.05% MMS for an additional 90 min. The cells were harvested and processed for immunoprecipitation, as mentioned above, except Protein-A conjugated dyna beads (Invitrogen, 10003D) were used instead of sepharose beads. 10 ng of DNA eluate was taken to prepare the NGS library using a TruSeq ChIP sample preparation kit from Illumina. The library was prepared as per the

manufacturer's protocol. Library QC was performed by measuring the concentration and size of the libraries by using the 4150 Tapestation system from Agilent and Qubit dsDNA HS assay kit (Thermo Fisher Scientific). The libraries were sequenced using the Illumina platform in paired-end mode with a read length of 151 bp by Macrogen Asia Pacific Pte. Ltd, Singapore.

## Bioinformatics analysis

The ChIP-Seq analysis was performed by Genotypic (Bengaluru, India). The raw reads were processed using FastQC (*Andrews, 2010*) for quality assessment and pre-processing, which includes removing the adapter sequences and low-quality bases (<q30) using TrimGalore (*Krueger, 2015*). The pre-processed high-quality read data was aligned to reference *Saccharomyces cerevisiae* genome database using Bowtie, an ultrafast, memory-efficient short read aligner geared towards quickly aligning large sets of short DNA sequences (reads) to large genomes. The aligned files (BAM files) were used for peak calling using MACS2 software with an input sample as a control. The bedgraph files generated from MACS2 software were used to plot the graphs using the IGV browser (*Robinson et al., 2011*). BEDTools (*Quinlan and Hall, 2010*) were used for further analysis and comparison of the peaks across the samples. MEME-ChIP (*Ma et al., 2014*) was utilized to identify the motifs present in the Scm3 binding sites. Further, the annotation of the peaks was made in comparison with the reference genome (r64: NCBI RefSeq: GCF_000146045.2). We also performed gene set functional enrichment using DAVID tools (*Sherman et al., 2022*).

## Protein extraction, gel running, and western blotting

Yeast cells harvested at the indicated time points were washed once with ice-cold water and once with ice-cold 20% TCA. To preserve the phosphorylation moieties, the cells were lysed in the presence of 1 X PhosSTOP (Roche). The pellet was resuspended in 200 µl of 20% TCA, an equal volume of glass beads was added, and cells were lysed using a mini bead beater (BIOSPEC, India) for 7 cycles (1 min ON, 2 min OFF on ice). The supernatant was collected after puncturing the O-ring tube and centrifuging at 2800 rpm for 2 min at 4 °C. 400 µl of 5% TCA was added to the beads and spun again to collect the supernatant. The total supernatant was spun at maximum speed for 10 min at 4 °C. The precipitate was kept for air-drying for 10–15 min before resuspending in 2 X SDS sample buffer and 1 X Tris (pH 8.0) for neutralization. Before loading, the protein was boiled for 5 min at 95 °C and spun at maximum speed for 30 s. Standard procedure was followed for SDS-PAGE and western blot transfer using PVDF membrane.

For the gel containing the PhosTag reagent, 10 mM $MnCl_2$ was added to the resolving gel along with 50 µM of the PhosTag reagent (AAL-107, FUJIFILM, JAPAN). Before western blotting, the gel was washed thrice with 1 mM EDTA for 10 min, followed by two washes in transfer buffer. The blot was incubated with primary antibodies at a dilution of 1:2500, while the secondary antibodies were used at 1:5000 dilution.

For phosphatase treatment, cell lysates were prepared in lysis buffer (25 mM HEPES, pH 7.5, 2 mM $MgCl_2$, 0.1 mM EDTA, 0.5 mM EGTA, 0.1 % NP-40, 15% glycerol, 80 mM NaF) supplemented with 1 X PMSF, 1 X PIC, 1 X Phos-Stop. The cells were lysed using 0.5 mm glass beads using a mini-bead beater (BIOSPEC, India) for 7 cycles (1 min ON, 2 min OFF on ice). After clarification of the lysate by centrifugation, 5 µg of anti-HA antibodies (ab9110, Abcam) was added and incubated at 4 °C overnight with gentle rotation followed by incubation with Protein G conjugated Dynabeads (10003D, Invitrogen) for 2 hr at 4 °C with rotation. The beads were washed thrice with lysis buffer followed by three times washing in phosphatase wash buffer (50 mM HEPES, pH 7.5; 0.2 mM $MnCl_2$, 100 mM NaCl, 5% glycerol, 2 mM β-mercaptoethanol). The eluate was resuspended in phosphatase buffer and incubated with or without FastAP (EF0651, Thermo) for 3 hr at 37 °C. After incubation, the proteins were eluted in 1 X SDS-PAGE buffer, samples were run on 12% SDS gel, and immunoblotting was performed using anti-HA antibodies.

To detect Scm3-6HA, Rad53, and γ-H2A, cell extracts were run on 12%, 8%, and 15% SDS gels, respectively. The proteins were transferred to PVDF membranes, which were cut to detect the above proteins and the control protein tubulin separately. To quantify the bands on the western blots, a

region of interest (ROI) was drawn around the band of interest, and the band's intensity was calculated using ImageJ. An ROI of the same size from a no-band area of the blot was used to calculate the background intensity. The background intensity was subtracted from the band intensity. Tubulin band intensities were measured in the same way. The intensity of the target bands (Scm3-6HA, Rad53, and γ-H2A) was divided by control tubulin band intensity to get the normalized values for the target bands, which were plotted using GraphPad Prism 9.0 (Version 9.4.1) software.

## Statistical analyses

The data presented in the main and figure supplements were obtained from two to three independent experiments. The error bars in the individual bar graphs represent the SEM. The statistical significance (p) was determined by a two-tailed Student's t-test, or one-way ANOVA test, as appropriate. A one-way ANOVA test was used when comparing more than one group; otherwise, a Student's t-test was used. The 'N' values denote the total number of cells analyzed from the combined 'n' number of replicates of the individual assays. The p values less than or equal to 0.05 are categorized as significant differences. The SD, SE, and statistical significance (p) values were calculated using GraphPad Prism 9.0 (Version 9.4.1) software.

## Acknowledgements

We acknowledge Prof. Martin Kupeic from Tel Aviv University, Israel and Dr. Saumitra Sau from Amity University, Kolkata, India for providing us with the yeast strains. This work is supported by the Department of Biotechnology (DBT), Govt of India, grant (BT/PR43050/BRB/10/1992/2021) to SKG. PA, AA, and SM are supported by fellowship grants from CSIR (09/087 (0972)/2019- EMR-I), DST (IF230484), and CSIR (09/0087 (17614)/2024-EMR-I), Govt of India, respectively. KH is supported by a PDF fellowship from IIT Bombay. We acknowledge the central instrumentation facility of IIT Bombay for Confocal Microscopes and Next-generation sequencing.

## Additional information

### Funding

| Funder | Grant reference number | Author |
| --- | --- | --- |
| Department of Biotechnology, Ministry of Science and Technology, India | BT/PR43050/ BRB/10/1992/2021 | Santanu K Ghosh |
| Council of Scientific and Industrial Research | 09/087 (0972)/2019- EMR-I | Prakhar Agarwal |
| Department of Science and Technology | IF230484 | Anushka Alekar |
| Council of Scientific and Industrial Research | 09/0087 (17614)/2024-EMR-I | Shubhomita Mallick |
| Indian Institute of Technology Bombay | | Kannan Harini |

The funders had no role in study design, data collection and interpretation, or the decision to submit the work for publication.

### Author contributions

Prakhar Agarwal, Conceptualization, Data curation, Formal analysis, Investigation, Visualization, Methodology, Writing – original draft, Writing – review and editing; Anushka Alekar, Data curation, Validation, Investigation, Visualization, Methodology; Shubhomita Mallick, Data curation, Formal analysis, Methodology; Kannan Harini, Data curation, Formal analysis, Investigation; Santanu K Ghosh, Conceptualization, Resources, Software, Formal analysis, Supervision, Funding acquisition, Validation, Investigation, Visualization, Methodology, Writing – original draft, Project administration, Writing – review and editing

## Author ORCIDs

Prakhar Agarwal ⬛ https://orcid.org/0000-0001-5794-545X
Anushka Alekar ⬛ https://orcid.org/0009-0002-8032-3397
Shubhomita Mallick ⬛ https://orcid.org/0009-0002-2169-7479
Kannan Harini ⬛ https://orcid.org/0000-0002-7792-2922
Santanu K Ghosh ⬛ https://orcid.org/0000-0002-3190-8084

## Decision letter and Author response

Decision letter https://doi.org/10.7554/eLife.104431.sa1
Author response https://doi.org/10.7554/eLife.104431.sa2

---

## Additional files

### Supplementary files

Supplementary file 1. Supplementary file related to main article file. (**A**) List of strains used in this study. (**B**) List of primers used in ChIP-qPCR.

MDAR checklist

### Data availability

All data generated or analysed during this study are included in the manuscript and supporting files.

The following dataset was generated:

| Author(s) | Year | Dataset title | Dataset URL | Database and Identifier |
|---|---|---|---|---|
| Agarwal P, Alekar A, Mallick S, Harini K, Ghosh SK | 2025 | Data from: Evidence of centromeric histone 3 chaperone involved in DNA damage repair pathway in budding yeast | https://doi.org/10.5061/dryad.gb5mkkx1t | Dryad Digital Repository, 10.5061/dryad.gb5mkkx1t |

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
