## [Editor Report]

DNA breaks occur frequently at centromeres and their repair is essential to maintain mitotic integrity in eukaryotes. The authors address this important question using budding yeast as a model system, with convincing data that demonstrates a role for the centromeric histone H3 chaperone Scm3 contributing to this repair pathway. These data have exciting implications for repair pathways that regulate an essential chromosomal locus required for mitosis and cell viability. The paper will be of interest to the centromere, chromosome instability and DNA repair fields.

---

## [Decision Letter]

[Editors' note: this paper was reviewed by Review Commons.]

Thank you for submitting your article "Evidence of centromeric histone 3 chaperone involved in DNA damage repair pathway" for consideration by *eLife*. Your article has been reviewed by 3 peer reviewers at Review Commons, and the evaluation at *eLife* has been overseen by Silke Hauf as the Senior Editor.

Based on your manuscript, the reviews and your responses, we invite you to submit a revised version incorporating the revisions as outlined in your response to the reviews.

When preparing your revisions, please also address the following points:

(1) For the reported Scm3 phosphorylation, please include a control where samples have been treated with phosphatase to demonstrate that observed shifts are due to phosphorylation.

(2) For Figure 5C, you conclude that there is no significant enrichment at the TUB2 locus (page 11), but this has not been statistically tested, and there does seem to be considerable enrichment at the 2-hour time point.

(3) For the ChIP experiment (Figure 3), are controls for the metaphase arrest available? Was the arrest similarly tight for the untreated and the MMS-treated sample?

(4) I cannot follow the argument about the enrichment of Scm3 after MMS treatment by ChIP (Figure 3). Almost all the peaks pointed out by black arrows are also observed in untreated cells. Please explain how this has been quantitatively assessed and provide additional metrics on all peaks observed without or with MMS treatment (more comprehensive than Table S1). How much of the variation observed is inter-experiment variability?

(5) In Figure 4, given the limited resolution of light microscopy, is it possible that any chromatin-associated protein would show an increased co-localization with Rad52 or γ-H2A (since the area of occupancy expands)? Please include a negative control, e.g. a transcription factor.

(6) When independent experiments were conducted (e.g. for Figure 2C/D, Figure 3B, Figure 4B/D, Figure 6E, S2B, etc.), please show the mean of each independent experiment, either in the main figure or in a separate supplemental figure. Statistical tests should additionally be conducted based on n = 3 experiments, not n > 300 cells.

(7) Please revise the title and incorporate "budding yeast" or "Scm3" into the title.

(8) Please try to shorten the discussion and avoid re-iterating results already covered in the Results section.

(9) Please be careful not to overstate the results. As one example: page 14 "Scm3… associates with the DDR proteins at the DNA damage sites." A direct protein-protein interaction has not been demonstrated. So please rephrase.

(10) In the abstract, line 24, "interacts with…" should be "genetically interacts with…".

(11) Line 203: Please rephrase "also referred to as the NLS". NLS is not a synonym for a bromodomain. Therefore, "..which also harbor an NLS" or "… which also harbor a candidate NLS" would be more appropriate.

(12) Please have the manuscript proofread for English grammar and spelling, in particular the introduction.

[Editors' note: further revisions were suggested prior to acceptance, as described below.]

Thank you for resubmitting the paper entitled "Evidence of centromeric histone 3 chaperone involved in DNA damage repair pathway in budding yeast" for further consideration by *eLife*. Your revised article has been evaluated by a Reviewing Editor and two of the original reviewers.

Two major remaining concerns were not sufficiently addressed by the revision:

(1) The claim that the DDR checkpoint is impaired in Scm3-depleted cells is not substantiated.

The new immunoblot in Figure 6A (formerly 6D) still shows no obvious reduction of Rad53 phosphorylation in Scm3-depleted cells.

Similarly, the new immunoblot for Pds1 in Figure 6E does not give any strong indication that Pds1 degrades earlier in Scm3-depleted cells. (Compare, for example: https://pmc.ncbi.nlm.nih.gov/articles/PMC312708/figure/F5/).

(2) The ChIP-seq experiment remains largely inconclusive regarding whether there is increased binding of Scm3 to chromatin after MMS treatment and whether there is enrichment at fragile sites.

The ChIP profiles from untreated and MMS-treated cells look highly similar, and it remains unclear whether differences in peak height between control and MMS are inter-experiment variability or MMS-induced changes. While a few Scm3 peaks overlap with fragile sites based on the new data you report, this is the case both in untreated and in MMS-treated cells. There is no clear increase after MMS treatment and in fact, some sites lose Scm3 enrichment after MMS treatment.

---

## [Author Response]

1. General Statements

We sincerely thank all three reviewers for their professional and constructive feedback. We appreciate the thorough evaluation of our manuscript and are committed to revising both the manuscript and supplemental materials based on the suggestions. We have carefully considered each comment and have addressed most of them in the initial revised version, which has been transferred. Additionally, we are currently conducting new experiments to provide the requested data to address a few comments. We are confident that these revision experiments will be completed in a couple of months or so, which will significantly enhance the quality of our study.

2. Description of the planned revisionsReviewer #1 (Evidence, reproducibility and clarity):In the manuscript by Agarwal and Ghosh, the authors examine yeast Scm3 function in the DNA damage response. They show that Scm3 loss results in DNA damage sensitivity and more Rad52 foci. Importantly, Scm3 is recruited to DSB sites using an HO-cut site and its loss results in an attenuated DNA damage checkpoint as measured by Rad53 phosphorylation. The authors demonstrate convincingly that Scm3, like its human counterpart HJURP, plays a role in the DNA damage response through altered Rad53 activation. However, what its specific role in DNA repair is remains ambiguous.Major comments:A. The phosphorylation aspect of Scm3 is intriguing and the authors show that Mec1 is not responsible for mediating its phosphorylation. Tel1 is another kinase that should be examined.

We thank the reviewer for the suggestion. We are in the process of examining the role of Tel1 kinase on Scm3 phosphorylation. The results from the experiment will be incorporated in the manuscript.

All other major and minor comments of Reviewer #1 have been addressed. Please see the section 3 below.

Reviewer #2 (Evidence, reproducibility and clarity):This manuscript describes a study implicating the Scm3 protein from budding yeast in the DNA damage response (DDR). Scm3 is a chaperone protein, whose main role is considered to be the loading of CENP-A(Cse4) at centromeres to facilitate chromosome segregation. However, the human ortholog of Scm3, HJURP, is known to have a role in DDR and in this study the authors provide evidence that Scm3 is also involved in the DDR in yeast. For the most part, the results presented support the conclusions made.

All the major and minor comments of Reviewer #2 have been addressed. Please see the section 3 below.

The western blots in the paper are not always entirely convincing. In addition, they are not described in enough detail to understand if a membrane was cut or if multiple gels were run. For example, the tubulin loading in Figure 6D is interrupted toward the end of the blot and the bands in Figure 7D go in different directions for the two blots for the MMS treated cells. In figure 6B, there are no detectable phospho-forms of Rad53 detected on the upper blot for the WT and scm3 lanes even though quantification is given on the right. It would be good to present better examples of the westerns or at least better describe what the reader is visualizing so the quantification and conclusions can be understood. How were the blots quantified? How were the westerns run and processed?

We have now included a separate paragraph in Materials and methods regarding the gel run, processing and quantification of the western blots in the revised manuscript for better understanding of the readers:

“To detect Scm3-6HA, Rad53, and g-H2A, the total proteins isolated from the appropriate cells were run on 12%, 8%, and 15% SDS gels, respectively. The proteins were transferred to the membranes, which were cut to detect the above proteins and the control protein tubulin separately. For the quantification of the bands on the western blots, a region of interest (ROI) was made around the band of interest, and the intensity of the band was calculated using ImageJ. A same ROI from a no-band area of the blot was used to calculate the background intensity. The background intensity was subtracted from the band intensity. The same process was done for the tubulin bands. The intensity of the target bands (Scm3-6HA, Rad53, and g-H2A) was divided by control tubulin band intensity to get the normalized values for the target bands, which were plotted using GraphPad Prism 9.0 (Version 9.4.1) software.” This has been added in page 25, lines: 881-890.”

Furthermore, we will again perform the experiments for a better representation of the western blots in figures 6B, D, and 7D.

The argument that scm3 depletion leads to a defect in DNA damage checkpoint activation is not strongly supported. Monitoring exit from the cell cycle by multibudding is not the most rigorous assay, especially since the image shows one cell with 5 nuclei. The authors should release cells from G1 into auxin and MMS and monitor cell cycle progression at least one other cell cycle marker, such as anaphase onset, DNA replication and/or Pds1 levels.

As per the reviewer’s suggestion, in order to support our argument that the absence of Scm3 causes a defect in DNA damage checkpoint activation, we will examine if these cells abrogate G2/M arrest and show an early anaphase onset. For this, we will monitor the levels of Pds1, as a marker of anaphase onset, along the cell cycle in wild type and Scm3-depleted cells both deleted for Mad2 to remove any inadvertent effect of spindle assembly checkpoint. The schematics of the experimental workflow is given in Author response image 1. Typically, the cells will be released from α factor arrest in the absence or presence of auxin (for the depletion of Scm3) and in the absence or presence of MMS. The samples will be harvested at the indicated time points and will be analyzed for:

Western blot: Pds1-Myc (to detect anaphase onset)Western blot: Rad53 and p-Rad53 (to detect DNA damage activation) (iii) Immunofluorescence: Tubulin (to detect cell cycle stages)

The results of the above experiment will be incorporated in the revised manuscript.

**Author response image 1. sa2fig1:** The schematic of the experimental strategy to be followed to elucidate the role of Scm3 in DNA damage checkpoint activation.

Minor comments:A. The ChIP-seq data is not publicly accessible. There is no reference to the data being available to review.

The data will be uploaded to the public domain.

All other major and minor comments of Reviewer #3 have been addressed. Please see the section 3 below.

3. Description of the revisions that have already been incorporated in the transferred manuscriptReviewer #1:Major comments:A. It is unusual to see multiple DNA repair foci as those observed in Figure 2B. What is the distribution of cells with 1, 2, 3, 4, or more foci? Are more observed in SCM3-AID cells perhaps suggesting that the DSB ends are not being clustered as would be expected in WT cells exposed to DNA damage?

As per the reviewer’s comment, we have included a graph (Author response image 2) showing the distribution of cells with 1, 2, 3, 4, or more Rad52-GFP foci when they are treated with MMS. There are more cells with 4 or more foci when Scm3 is depleted (*SCM3-AID* + Auxin) compared to the wild type (*SCM3-AID*). The average number of Rad52-GFP foci per cell presented in Author response image 2 (2.8 in the mutant vs. 1.9 in the wild type) is well in accord with the previous report (Conde and San-Segundo, 2008), where the same was reported as ~2.5 in the cells lacking a methyl transferase Dot1, vs ~ 1.5 in the wild type. More Rad52-GFP foci in MMS-treated cells lacking Scm3 may arise due to the creation of too many damaged sites to be accommodated in 1-2 foci and/or due to the inability of the cells to cluster the DSB ends.

**Author response image 2. sa2fig2:** The percentage of wild type (*SCM3-AID*) and Scm3 depleted (*SCM3-AID*+Auxin) cells harboring indicated numbers of Rad52-GFP foci upon treatment with 0. 02% MMS.

This result has been incorporated as a new supplementary Figure S4C and new text has been added in the revised manuscript as: “We further quantified the distribution of cells with 1, 2, 3, or >4 Rad52-GFP foci in wild type (*SCM3-AID*) or Scm3 depleted (*SCM3-AID* + auxin) cells treated with MMS. Scm3 depleted cells showed a significantly higher number of cells with more than >4 Rad52-GFP foci, suggesting the possibility of the creation of too many damaged sites to be accommodated in 1-2 foci or the inability of such cells to cluster the DSB ends.” in page 7, lines: 237-241.

B. The peaks with increased Scm3 recruitment by ChIP-seq upon MMS is confusing as MMS does not induce specific damage at genomic locations. Is Scm3 being recruited at other genomic sites that might be more susceptible to DNA damage? Is Scm3 recruited to Pol2 sites for example? Or fragile sites?

We believe that the MMS induced increase in association of Scm3 with the noncentromeric chromatin loci depends on MMS sensitive vulnerable chromosomal sites. We agree with the reviewer that MMS might cause DNA damage at these sites, leading to Scm3 occupancy at those sites. Therefore, we compared the sites of Scm3 occupancy with possible such sites available from the literature that include fragile sites, RNA Pol II binding sites, double strand break hotspots, and coldspots. Based on our analysis, we have included the following lines in the ‘discussion’ section in page 16-17, lines 566-594 as follows:

“Moreover, an overall increase in the chromatin association of Scm3 in response to MMS also suggests that Scm3 might be recruited to several repair centers or sites that are susceptible to DNA damage, for example, the fragile sites (Figure 3B, C, E, S6). These sites in yeast are DNA regions prone to breakage under replication stress, often corresponding to replication-slow zones (RSZs) (Lemoine et al., 2005). These regions include replication termination (TER) sequences, tRNA genes, long-terminal repeats (LTRs), highly transcribed genes, inverted repeats/palindromes, centromeres, autonomously replicating sequences (ARS), telomeres, and rDNA (Song et al., 2014). Since the helicase Rrm3 is often associated with these fragile regions (Song et al., 2014), we compared Scm3 binding sites with the top 25 Rrm3 binding sites from the literature (Azvolinsky et al., 2009). In untreated cells, Scm3 sites overlapped with three Rrm3 sites on chromosomes X, XII, and XIV. Whereas in MMS treated cells, overlapping was found with four Rrm3 sites, with two (on chromosomes XII and XIV) shared with untreated cells and two new sites were observed on chromosomes II and XII (Table R1). Mapping of the Scm3 sites with the tRNA genes and LTRs revealed that these sites from the untreated cells did not overlap with the LTRs (Raveendranathan et al., 2006). However, the same from the treated cells showed overlap with two LTRs on chromosome XVI. No overlap with tRNA genes was observed in the treated cells (Table R1). We next examined Scm3 occupancy at 71 TERs documented in the literature (Fachinetti et al., 2010). Scm3 was found to bind to 6 TERs in both untreated and MMS-treated cells. Notably, MMS treatment resulted in three new peaks, while three peaks were shared with untreated samples (Table R1). Lastly, we compared Scm3 sites with top 25 RNA Pol II sites obtained from the literature (Azvolinsky et al., 2009). In untreated cells, Scm3 was found at only one of these Pol II sites, whereas after MMS treatment, Scm3 sites overlapped with four such sites (Table R1). We further checked the occupancy of Scm3 at a few DSB hotspots (*BUD23*, *ECM3*, and *CCT6*) and DSB coldspot (*YCR093W*) as mentioned in the literature (Dash et al., 2024; Nandanan et al., 2021). However, we did not find Scm3 binding to these sites. Overall, in-silico analysis of the binding sites indicates that the non-centromeric enrichment of Scm3 occurs at sites that are amenable to DNA damage.”

**Author response table 1. sa2table1:** The table summarising the occupancy of Scm3 in untreated or MMS treated conditions at the indicated regions.

Region	Chromosome	Scm3 occupancy	
		**Untreated**	**MMS treated**
Rrm3 bindingsites	Chr II		YES
	Chr X	YES	
	Chr XII		YES
	Chr XII	YES	YES
	Chr XIV	YES	YES
LTRs	Chr XVI		YES
	Chr XVI		YES
tRNA	Chr XV	YES	
TERs	Chr IV	YES	
	Chr V		YES
	Chr VI	YES	YES
	Chr VII		YES
	Chr X		YES
	Chr X	YES	
	Chr XIV	YES	YES
	Chr XV	YES	YES
	Chr XVI	YES	
Pol II binding sites	Chr II		YES
	Chr X	YES	
	Chr XII		YES
	Chr XII		YES
	Chr XV		YES

It is hard to see what MMS resistance the authors state is observed in Mif2-depleted cells in Figure S3. Perhaps this could be better explained or the claim removed.

We agree with the comment and have removed the claim from the manuscript.

Protein levels of Scm3 or any of the other factors depleted with AID were never assessed.

We have assessed the protein level of Scm3 and a control protein, tubulin using western blotting as per the reviewer’s suggestion (Figure1—figure supplement 2). We did not observe any significant change in the protein levels in *SCM3-HA* or *SCM3-HA-AID* cells, suggesting that the AID tagging of Scm3 per se did not make the cells non-functional and the protein was degraded as expected upon addition of auxin. Moreover, the *SCM3-AID* cells were used previously to examine the effect of Scm3 on kinetochore assembly (Lang et al., 2018).

This result has been incorporated as Figure1—figure supplement 2C, and new text has been added in the revised manuscript as: “The depletion of Scm3 was verified by observing a higher percentage of G2/M arrested cells and by Western blot verifying the degradation of Scm3-AID after auxin treatment (Figure 1—figure supplement 2B-C).” in page 5, lines: 142-144.

Reviewer #2 (Evidence, reproducibility and clarity (Required)):This manuscript describes a study implicating the Scm3 protein from budding yeast in the DNA damage response (DDR). Scm3 is a chaperone protein, whose main role is considered to be the loading of CENP-A(Cse4) at centromeres to facilitate chromosome segregation. However, the human ortholog of Scm3, HJURP, is known to have a role in DDR and in this study the authors provide evidence that Scm3 is also involved in the DDR in yeast.For the most part, the results presented support the conclusions made.Major comments:A. Figure 1 Could depletion of Scm3 arrest cells in late G2/M and it is this delay that causes increased sensitivity to DNA damaging agents? A control with nocodazole or other means – that also arrests cells at this point – might provide a nice control for this. Perhaps the other kinetochore mutants, used therein, achieve this control – but cell cycle phase would need to be assessed.

We thank the reviewer for pointing out to a probable effect of the cell cycle stage on the observed MMS sensitivity. In fact, we were also concerned that the observed DNA damage sensitivity in Scm3 depleted cells might be due to G2/M arrest. To rule out this possibility, we monitored Rad52-GFP foci as a marker for DNA damage in the wild type and Scm3 depleted cells both arrested at G2/M using nocodazole (Figure 2—figure supplement 2). While Scm3 depleted condition exhibited >20% Rad52-GFP positive cells, less than 10% wild type cells showed the same in the absence of any DNA damaging agents (Figure 2—figure supplement 2C, no MMS, 60 mins). Upon challenging these cells with MMS in the presence of nocodazole, Scm3 depleted condition exhibited over 40% Rad52-GFP positive cells, whereas less than 20% wild-type cells harboured Rad52-GFP. This significant increase in Rad52-GFP positive cells when Scm3 is depleted clearly indicates that the observed MMS sensitivity in these cells is due to the absence of Scm3 rather than due to an effect of a cell cycle stage. Furthermore, we have also used Cdc20 depleted G2/M arrested cells as a wild type control to test the activation of the DNA damage checkpoint by Rad53 phosphorylation. These cells showed robust Rad53 activation in response to MMS, in contrast to poor activation in Scm3 depleted cells (Figure 6), suggesting that G2/M arrest is not the reason for the DNA damage sensitivity observed in the latter cells.

However, as per the reviewer's suggestion, we examined the MMS sensitivity of the wild type cells arrested at G2/M by nocodazole. As expected, these cells did not show increased sensitivity which further confirms that the DNA damage sensitivity observed in the *scm3* mutant is not due to G2/M arrest. This result has been incorporated within Figure 1—figure supplement 3B, replacing the earlier figure.

To include this result, we have included new text, and revised the result section in page 5-6, lines 158-179 as follows:

“The increased sensitivity of *scm3*-depleted cells to DNA-damaging agents could be due to the weakening of the kinetochores as Scm3-mediated deposition of Cse4 promotes kinetochore assembly or due to the delay in cell cycle, as Scm3 depleted cells arrest in late G2/M phase (Camahort et al., 2007; Cho and Harrison, 2011). If either of these holds true, perturbation of the kinetochore by degradation of other kinetochore proteins or wild type cells arrested at metaphase must show a similar sensitivity to MMS. In budding yeast, Ndc10 is recruited to the centromeres upstream of Scm3 (Lang et al., 2018), whereas the centromeric localization of Mif2, another essential inner kinetochore protein, depends on Scm3 and Cse4 (Xiao et al., 2017). We constructed *NDC10-AID* and *MIF2-AID* strains and used them for our assay to represent the proteins independent or dependent on Scm3 for centromeric localization, respectively. We also included one non-essential kinetochore protein, Ctf19, a protein of the COMA complex, to remove any possible mis-judgement in distinguishing cell-growth-arrest phenotype occurring due to drug-sensitivity vs. auxin-mediated degradation of essential proteins. The COMA complex is directly recruited to the centromeres through interaction with the N terminal tail of Cse4, hence dependent on Scm3 (Chen et al., 2000; Fischböck-Halwachs et al., 2019). Mid-log phase cells were harvested and spotted on the indicated plates, however, we did not observe any increased sensitivity of such cells to MMS (Figure 1—figure supplement 3). Further, wild type cells, when challenged in the presence of nocodazole and MMS, also did not show any increased sensitivity to MMS. Therefore, the increased sensitivity to MMS in *scm3* mutant but not in other kinetochore mutant or metaphase arrested cells indicates that Scm3 possesses an additional function in genome stability besides its role in kinetochore assembly.”

Further we have also revised the Discussion section to include the observed results in page 15-16, lines 535-539 as follows:

“The observed sensitivity is not due to problems in kinetochore formation or cell cycle progression in the absence of Scm3, as other kinetochore mutants or metaphase block in wild type cells did not show similar sensitivity (Figure 1—figure supplement 3), indicating that the drug sensitivity phenotype of Scm3 depleted cells is not due to weakly formed kinetochores or cell cycle delay.”

Mutants of the HR pathway in yeast (e.g. rad52∆ with mre11∆ for example) are typically epistatic. The observation that Scm3 depletion is not epistatic with rad52∆ (Figure 1C) suggests the Scm3 acts via another pathway than the classic Rad52 HR pathway. This should be pointed out and discussed.

We have now included the discussion “In yeast, although HR is the preferred repair pathway, in the case of perturbed HR, an alternate pathway named non-homologous end joining (NHEJ) can occur. The absence of epistatic interaction between *SCM3* and *RAD52* (Figure 1C) suggests that Scm3 may function in ways other than the Rad52-mediated classical HR pathway. In this context, it would be interesting to test how Scm3 might interact with the key proteins of the NHEJ pathway, such as Ku70/Ku80 and Lig4 (Gao et al., 2016). It is possible that Scm3 may promote a certain chromatin architecture facilitating the DSB ends to stay together to be accessible for NHEJ-mediated end joining.” in page 16, lines 541-548.

Figure 2 should include auxin treatment of RAD52-GFP cells (without the Scm3 degron) to show that the auxin treatment alone does not increase Rad52 foci.

We performed the suggested experiment and did not observe any significant increase in Rad52-GFP positive cells when treated the cells with auxin+DMSO as compared to only DMSO.

This result has been incorporated as a new supplementary Figure (Figure 2—figure supplement 1A-B) and new text has been added in the revised manuscript as “To rule out the possibility that auxin treatment alone can cause increased Rad52-GFP foci formation, we challenged the wild type (*RAD52-GFP*) cells with auxin or DMSO and counted the number of cells with Rad52-GFP foci. We did not observe any increase in Rad52-GFP positive cells when treated with auxin+DMSO as compared to only DMSO (Figure 2—figure supplement 1A-B).” in page 7, lines: 231-235.

Line 246-247 For the data presented, it seems to me possible that Scm3 depleted cells may indicate a defective DDR pathway (as stated) or may indicate defects in DNA replication or an increase in some other form of DNA damage?

We agree with the reviewer’s comment that the depletion of Scm3 can cause replication error or other form of DNA damage in addition to the defect in DDR pathway. To include this, we have modified the sentence as “Taken together, Scm3 depleted cells exhibit more Rad52 foci, indicating a compromised DDR pathway in these cells. However, defects in DNA replication or creation of other DNA lesions producing more foci also cannot be ruled out.” in page 8, lines 254-256.

In Figure 1 and throughout, please describe in the figure legends how error bars and p values are derived, and the number of experiments involved.

We have now verified all the figure legends and described how error bars and p values are derived and have mentioned the number of experiments involved.

Line 35 replace 'cell survival' with 'cell division' – non-dividing cells can survive fine without chromosome segregation. See also line 62.

We have now changed ‘cell survival’ with ‘cell division’ in lines 35 and 62.

Line 52 and throughout, I suggest replacing CenH3 with CENP-A or Cse4. The term CenH3 is confusing since regional centromeres contain both CENP-A nucleosomes and H3 nucleosomes – the latter of which can also be called CenH3 nucleosomes.

We have replaced CenH3 with CENP-A or Cse4 at the appropriate locations.

Lines 69-79 specific references are needed for the sentences starting "HJURP was so named…", "In addition,…", "As a corollary,…" and "Finally,…" The final sentence of this paragraph, starting "Perhaps due to…" is unclear.

We have included the reference as mentioned by the reviewer. Also, we have changed the last line as “Notably, HJURP has been visualized to be diffusely present throughout the nucleus (Dunleavy et al., 2009; Kato et al., 2007), possibly due to its global chromatin binding and involvement in DDR.” in page 3, lines 74-76.

Line 96 "gross chromatin" is unclear; also line 476.

We have changed gross chromatin to “bulk of the chromatin.” and incorporated it into the main text.

Line 103 "dimerize"

We have replaced ‘dimerizes’ with ‘dimerize’.

Line 109 "most" and "highly" don't work together – perhaps better to say "the functions appear conserved from humans to yeast".

have changed the wording as the reviewer suggested.

Line 175 "grown" to "phase", see also line 223.

We have changed the wording as the reviewer suggested.

Line 293 delete "besides"

We have deleted the word ‘besides’.

Figure 5 – panels C and D, please make x axis labels clearer – they are directly underneath the 2kb ChIP. They should include a horizontal bar to indicate that all 5 ChIP experiments are included in each time point.

We have now included a horizontal bar in both Figure 5 and the corresponding supplementary Figure 5—figure supplement 1, to better represent the ChIP experiments. We thank the reviewer for pointing this out.

Reviewer #3Major comments:A. The quantification in Figure 3B is not clear. Is it done on a per/nuclei basis? What pools of Scm3 and Ndc10 are being normalized?

We have now included a separate paragraph in Materials and methods regarding the gel run, processing and quantification of the western blots in the revised manuscript for better understanding of the readers:

“To detect Scm3-6HA, Rad53, and γ-H2A, the total proteins isolated from the appropriate cells were run on 12%, 8%, and 15% SDS gels, respectively. The proteins were transferred to the membranes, which were cut to detect the above proteins and the control protein tubulin separately. For the quantification of the bands on the western blots, a region of interest (ROI) was made around the band of interest, and the intensity of the band was calculated using ImageJ. A same ROI from a no-band area of the blot was used to calculate the background intensity. The background intensity was subtracted from the band intensity. The same process was done for the tubulin bands. The intensity of the target bands (Scm3-6HA, Rad53, and γ-H2A) was divided by control tubulin band intensity to get the normalized values for the target bands, which were plotted using GraphPad Prism 9.0 (Version 9.4.1) software.” This has been added in page 25, lines: 890-899.

We have now replaced Figure 6D with Figure 6B with a better blot representing the reduced Rad53 phosphorylation in *scm3* as compared to *cdc20*. For Figure 7D, we have reprocessed the image and also deposited uncropped and unprocessed source blots depicting different blots that were run to detect Scm3 and tubulin.

Line 103: not clear what "both the proteins dimerize" means…probably should be "both proteins dimerize"

We have changed the wording to “both proteins dimerize”.

The argument that Ndc10 does not have a growth defect on MMS is a weak conclusion, given that almost no control cells grow on auxin in the absence of MMS.

We have now repeated the spotting assay with a lesser concentration of auxin and replaced Figure S3 with a new Figure 1—figure supplement 3 to better represent and conclude that the loss of Ndc10 does not cause MMS sensitivity.

4. Description of analyses that authors prefer not to carry outReviewer #3Minor comments:A. The model is elegant but there are chromatin pools (beyond the kinetochore pool) of Scm3 that do not contain Rad52 and/or γ-H2X and vice versa. It would be helpful if the authors could speculate on how to reconcile these different pools. It might be premature to suggest such a detailed model at this point since the function of Scm3 in the checkpoint is still very unclear so I would encourage the authors to make a less detailed model.

By showing the green hallow, we have depicted the nuclear pool of Scm3, and we have not shown that the pool contains DDR proteins viz., Rad52 or g-H2A. Rather, we have shown the recruitment of these proteins at the DNA damage sites. Since the focus of this manuscript is on the non-centromeric functions of Scm3, we have not shown the kinetochore pool of Scm3. Although the model is a detailed one, the contribution from this work has been mentioned legitimately at every stage so that the readers can judge the merit of this work. We believe that a detailed model would provide a better perspective to the readers to correlate the revealed as well as yet-to-reveal functions of Scm3 in a spatiotemporal manner with the other players of the DDR pathway. Therefore, we prefer to keep the model in a detailed form.

[Editors’ note: what follows is the authors’ response to the second round of review.]

Based on your manuscript, the reviews and your responses, we invite you to submit a revised version incorporating the revisions as outlined in your response to the reviews.When preparing your revisions, please also address the following points:(1) For the reported Scm3 phosphorylation, please include a control where samples have been treated with phosphatase to demonstrate that observed shifts are due to phosphorylation.

We thank the editor for the suggestion. We have now performed a phosphatase assay demonstrating the shifts are indeed due to Scm3 phosphorylation. Figure 7—figure supplement 1F shows the disappearance of the Scm3 phosphorylated species (Scm3*) after treatment with alkaline phosphatase.

We have included this in the manuscript in Figure 7—figure supplement 1 and incorporated in the text as follows on page 15 lines 521-525:

“We further verified this by performing a phosphatase experiment. *SCM3-6HA* cells treated or untreated with MMS were immunoprecipitated and subjected to alkaline phosphatase treatment for 3 hrs at 37°C. The disappearance of the Scm3* band upon phosphatase treatment confirms that the shift was indeed due to phosphorylation (Figure 7—figure supplement 1F).”

We have now also included this in the Materials and methods section as follows on pages 25-26 lines 897-909 as follows:

“For phosphatase treatment, cell lysates were prepared in lysis buffer (25 mM HEPES, pH7.5, 2 mM MgCl_2_, 0.1 mM EDTA, 0.5 mM EGTA, 0.1 % NP-40, 15 % glycerol, 80 mM NaF) supplemented with 1X PMSF, 1X PIC, 1X Phos-Stop. The cells were lysed using 0.5 mm glass beads using a mini-bead beater (BIOSPEC, India) for 7 cycles (1 min ON, 2 mins OFF on ice). After clarification of the lysate by centrifugation, 5 µg of anti-HA antibodies (ab9110, Abcam) was added and incubated at 4°C overnight with gentle rotation followed by incubation with Protein G conjugated Dynabeads (10003D, Invitrogen) for 2 hrs at 4°C with rotation. The beads were washed thrice with lysis buffer followed by three times washing in phosphatase wash buffer (50 mM HEPES, pH 7.5; 0.2 mM MnCl_2_, 100 mM NaCl, 5% glycerol, 2 mM β-mercaptoethanol). The eluate was resuspended in phosphatase buffer and incubated with or without FastAP (EF0651, Thermo) for 3 hrs at 37°C. After incubation, the proteins were eluted in 1X SDS-PAGE buffer, samples were run on 12% SDS gel, and immunoblotting was performed using anti-HA antibodies.”

(2) For Figure 5C, you conclude that there is no significant enrichment at the TUB2 locus (page 11), but this has not been statistically tested, and there does seem to be considerable enrichment at the 2-hour time point.

In Figure 5C, the statistical analyses were given in a pairwise fashion between each locus and the *TUB2* locus. At the *TUB2* locus, no significant enrichment of Scm3-6HA was found at different time points of HO induction (Figure 5—figure supplement 1). However, we have now performed a separate statistical analysis for the enrichment of Scm3 at both *CEN3* and *TUB2* loci (Figure R2) and incorporated the results in the text on page 12 lines 411-416 as follows:

“To confirm that the enrichment of Scm3 at 2 hrs at the HO-induced DSB sites is a site-specific response rather than a global chromatin enrichment, we verified the enrichment of Scm3-6HA at the *CEN3* and *TUB2* loci at different time points of HO induction. We could not observe any significant enrichment of Scm3 at those loci over different time points, suggesting Scm3 specifically enriches at the DSB sites upon DNA damage (Figure 5—figure supplement 1D, E).”

(3) For the ChIP experiment (Figure 3), are controls for the metaphase arrest available? Was the arrest similarly tight for the untreated and the MMS-treated sample?

The samples were first analysed for metaphase arrest by DAPI staining before proceeding with ChIP. Figure 3—figure supplement 1A shows that both MMS treated and untreated cells were similarly arrested at the metaphase stage. We are thankful to the editor for this comment.

We have included this new supplementary figure: Figure 3—figure supplement 1A and incorporated in the text as follows on page 8 lines 282-283.

“Both the MMS treated and untreated samples were similarly arrested as judged by DAPI staining prior to ChIP-seq assay (Figure 3—figure supplement 1A).”

(4) I cannot follow the argument about the enrichment of Scm3 after MMS treatment by ChIP (Figure 3). Almost all the peaks pointed out by black arrows are also observed in untreated cells. Please explain how this has been quantitatively assessed and provide additional metrics on all peaks observed without or with MMS treatment (more comprehensive than Table S1). How much of the variation observed is inter-experiment variability?

We thank the editor for the comment and agree that the ChIP-seq data presentation in its current form might be confusing. We have now provided a revised version of Figure 3 with a better representation of the Scm3-binding peaks by using distinct and different colored bars that explain the ChIP-seq data in a much better way. We have quantitatively analysed the peaks obtained in the untreated and treated samples, and only those sites where the enrichment upon immunoprecipitation (in the treated or untreated sample) exceeds that of the input by more than one-fold are considered as the true peaks. Upon quantification from MACS2 results, we observed that in the untreated samples, 17 peaks were enriched (~1.4-1.8 fold) compared to the input, while 75 peaks were enriched (~1.3-1.9 fold) in the MMS treated samples at the non-centromeric sites (Figure 3—figure supplement 1B, Figure 3D). Out of these peaks, two were commonly found in both samples. One peak was found in the *FLO1* gene of Chr II, while the other peak was found in the CT repeat region of the indole pyruvate decarboxylase1 gene of Chr XII. Interestingly, as commented by the reviewer, we observed ~1.4 fold enrichment of Scm3 binding at the rDNA, that harbors repetitive regions, in the treated cells (1.54 fold enrichment relative to input) compared to untreated cells (1.06 fold enrichment relative to input) (Figure 3E) suggesting its general preference towards such regions. This analysis clearly demonstrates an increase in the binding of Scm3 to the non-centromeric sites upon MMS treatment.

To further understand the nature of the non-centromeric sites harboring the enriched peaks (17 untreated, 75 treated) they were scanned for the presence of any motifs, and we observed three and one significant motifs (Figure 3—figure supplement 1F) in the treated and in the untreated samples, respectively. The three motifs in the MMS treated samples were specifically observed in the viral LTR region of the gag-pol fusion gene, based on the annotation from the NCBI reference genome (NCBI accession: GCF_000146045.2). These viral LTRs are found in multiple chromosomes spanning 29 peaks (~39% of total peaks) in the treated samples. Interestingly, these regions are considered fragile sites marked by the binding of Rrm3 helicase (Song et al., 2014). Notably, these viral LTRs are also known to trigger DNA-damage response pathways (Sinclair et al., 2006). This suggests that upon DNA damage induction, Scm3 is recruited to genomic locations that are more prone to damage, perhaps by interaction with DNA damage repair proteins. This can be argued based on the increased co-localization of Scm3 with DNA damage repair proteins Rad52, and y-H2A (Figure 4 of the manuscript) and the recruitment of Scm3 to the HO-induced DSB site (Figure 5 of the manuscript), where Scm3 binds DNA up to -3 kb from the DNA cut site, and its enrichment correlates with the binding of Rad51, a DNA damage repair protein.

Additionally, we also scanned the peaks for different features and observed that MMS treated peaks comprise fragile sites (Song et al., 2014) and hotspots (Gerton et al., 2004), which are prone to DNA damage. The untreated samples had a comparatively smaller number of such damage-prone sites (Figure 3—figure supplement 1C). Moreover, the MMS treated samples had a higher number of peaks with short tandem repeats (length 5-7 nucleotides), interspersed repeats and low complexity regions (less diversity of nucleotides). These repeat regions are also shown to be a source of DNA damage, as they form alternative non-B DNA structures, which are more prone to damage (Brown et al., 2021; Wierdl et al., 1997). Further gene set enrichment analysis of the 17 non centromeric peaks in untreated cells revealed that Scm3 binds to regions that are involved in normal metabolic pathways such as glycolysis/gluconeogenesis, regulation of amino acids, and telomere maintenance (Figure 3—figure supplement 1E) while in the treated samples the 75 non-centromeric peaks had GO terms/pathways that are mostly related to nucleotide synthesis and detoxification/flocculation, which are usually triggered in stress conditions (Figure 3—figure supplement 1D). We speculate that Scm3 might bind to these genes to ameliorate the DNA damage stress caused by MMS treatment. Hence, we could attribute the increased binding of the Scm3 to the non-centromeric regions to DNA damage/repair.

Based on our analysis, we have edited the manuscript as follows on pages 9-10, lines 287-329:

“The Scm3 ChIP-seq signal was normalised with the input signal, and upon quantification from MACS2, we observed the enrichment (peaks) of Scm3 at all 16 centromeres in both untreated and MMS treated samples (Figure 3D, blue bars). Scm3 was also enriched at the 9.1 kb rDNA region on Chr XII in both untreated and treated samples (Figure 3E, at rDNA), consistent with our chromatin spread data (Figure 3A). Upon further analysis, the enrichment of Scm3 at rDNA, which harbours repetitive regions, was found to be ~1.4-fold in the treated cells (1.54-fold enrichment relative to input) compared to untreated cells (1.06-fold enrichment relative to input) (Figure 3E), suggesting its general preference towards such regions.

Besides the centromeric peaks, 17 non-centromeric peaks (~1.4–1.8-fold enrichment over input) were detected in untreated samples, compared to 75 peaks (~1.3–1.9-fold enrichment over input) in MMS-treated cells, in which only two peaks were shared (Figure 3—figure supplement 1B). This analysis clearly demonstrates an increase in the binding of Scm3 to the non-centromeric sites upon MMS treatment. The feature analysis of these sites revealed that MMS treated samples were enriched in peaks which comprise fragile sites (Song et al., 2014) and recombination hotspots (Gerton et al., 2004), which are prone to DNA damage. The untreated samples had a comparatively smaller number of such damage-prone sites (Figure 3—figure supplement 1C, fragile sites and hotspots). Moreover, the MMS treated samples had a higher number of peaks with short tandem repeats (length 5-7 nucleotides), interspersed repeats and low complexity regions (less diversity of nucleotides) than in the untreated samples. These repeat regions are also shown to be a source of DNA damage, as they form alternative non-B DNA structures, which are more prone to damage (Brown et al., 2021; Wierdl et al., 1996). Furthermore, analysis of the gene ontology (GO) associated with the 17 non-centromeric peaks revealed GO terms that are involved in normal metabolic pathways such as glycolysis/gluconeogenesis, regulation of amino acids, and telomere maintenance (Figure 3—figure supplement 1E) while in the treated samples the 75 non-centromeric peaks had GO terms that are mostly related to nucleotide synthesis and detoxification/flocculation, which are usually triggered in stress conditions (Figure 3—figure supplement 1D).We speculate that Scm3 might bind to these genes to ameliorate the DNA damage stress caused by MMS treatment. Hence, we could attribute the increased binding of the Scm3 to the non-centromeric regions to DNA damage.

To further understand the nature of the non-centromeric sites harboring the enriched peaks (17 untreated, 75 treated), they were scanned for the presence of any motifs, and we observed three and one significant motifs in the treated and in the untreated samples, respectively (Figure 3—figure supplement 1F). The three motifs in the MMS treated samples were specifically observed in the viral LTR region of the gag-pol fusion gene, based on the annotation from the NCBI reference genome (NCBI accession: GCF_000146045.2). These viral LTRs are found in multiple chromosomes spanning 29 peaks (~39% of total peaks) in the treated samples. Interestingly, these regions are considered fragile sites marked by the binding of Rrm3 helicase (Song et al., 2014). Notably, these viral LTRs are also known to trigger DNA-damage response pathways (Sinclair et al., 2006). This suggests that upon DNA damage induction, Scm3 is recruited to genomic locations that are more prone to damage, perhaps by interaction with DNA damage repair proteins.”

(5) In Figure 4, given the limited resolution of light microscopy, is it possible that any chromatin-associated protein would show an increased co-localization with Rad52 or γ-H2A (since the area of occupancy expands)? Please include a negative control, e.g. a transcription factor.

We are thankful to the editor for pointing to include the negative control. In the revised version, we have included Hsf1, a transcription factor, as a negative control in the co-localization assay. We examined co-localisation between y-H2A and Hsf1-13Myc and unlike Scm3-6HA, found no increased co-localisation in response to MMS or heat shock between the two proteins (Figure 4—figure supplement 2).

The result has been incorporated in the text as follows on pages 10-11 lines 346-356:

“In order to negate the possibility that the increased co-localisation of Scm3-6HA with Rad52 or γ-H2A following MMS treatment is merely due to the increased accumulation of the DDR proteins in response to MMS within the small yeast nucleus, we tested the colocalization of an unrelated transcription factor, Hsf1, with γ-H2A. Since Hsf1 is known to increase its chromatin association in response to heat shock (Chowdhary et al., 2019; Rubio et al., 2024), we verified this by incubating the cells at 37°C for 15 mins. As expected, Hsf1 showed an increased chromatin association (a greater number of Hsf1-13Myc foci) at high temperatures compared to normal temperatures (Figure 4—figure supplement 2A-C). As expected, the increased chromatin occupancy of Hsf1 or γ-H2A due to high temperature or MMS treatment, respectively, did not result in any significant increase in the co-localization frequency between the two proteins (Figure 4—figure supplement 2D-F).”

We further verified if the assay was sensitive enough and does not show an increased co-localisation due to the limited resolution of light microscopy. For clarity of the experiment, we recalculated the PCC values by rotating the red image 180° as performed in our earlier studies (Ma et al., 2023). As per the analysis, we did not observe any significant increase in the PCC values after the MMS treatment, suggesting that the observed values (Figure 4D) are not an artefact due to experimental errors (Figure 4—figure supplement 1). We have included these results in the manuscript on page 11 lines 359-364 as follows:

“We verified the accuracy of the assay by rotating the red and green fluorescence images through 180° relative to each other and recalculating the PCC value as described earlier (Ma et al., 2023). After rotation, we failed to observe any significant correlation between Scm3-13Myc and γ-H2A (Figure 4—figure supplement 1G-I), suggesting the observed high PCC values (Figure 4D-F) are not an artefact of the assay.”

(6) When independent experiments were conducted (e.g. for Figure 2C/D, Figure 3B, Figure 4B/D, Figure 6E, S2B, etc.), please show the mean of each independent experiment, either in the main figure or in a separate supplemental figure. Statistical tests should additionally be conducted based on n = 3 experiments, not n > 300 cells.

We have now edited all the figures to represent the mean of each experimental replicate.

(7) Please revise the title and incorporate "budding yeast" or "Scm3" into the title.

We have revised the title to “Evidence of centromeric histone 3 chaperone involved in DNA damage repair pathway in budding yeast.”

(8) Please try to shorten the discussion and avoid re-iterating results already covered in the Results section.

We have now shortened the discussion as per the suggestion.

(9) Please be careful not to overstate the results. As one example: page 14 "Scm3… associates with the DDR proteins at the DNA damage sites." A direct protein-protein interaction has not been demonstrated. So please rephrase.

It has been reworded to “Scm3 possibly associates with the DDR proteins at the DNA damage sites”.

(10) In the abstract, line 24, "interacts with…" should be "genetically interacts with…".

We have made the necessary changes.

(11) Line 203: Please rephrase "also referred to as the NLS". NLS is not a synonym for a bromodomain. Therefore, "..which also harbor an NLS" or "… which also harbor a candidate NLS" would be more appropriate.

The statement has been modified to “which also harbors a candidate NLS.”

(12) Please have the manuscript proofread for English grammar and spelling, in particular the introduction.

We have now corrected all the grammatical errors.

[Editors’ note: what follows is the authors’ response to the third round of review.]

Two major remaining concerns were not sufficiently addressed by the revision:(1) The claim that the DDR checkpoint is impaired in Scm3-depleted cells is not substantiated.The new immunoblot in Figure 6A (formerly 6D) still shows no obvious reduction of Rad53 phosphorylation in Scm3-depleted cells.

The quantification of perturbed Rad53 phosphorylation is shown in Figure 6—figure supplement 1B, C, where in Scm3-depleted cells, the phosphorylation of Rad53 (activation of DDR checkpoint) is significantly reduced. This has been further verified in Figure 6A (new immunoblot). The quantification of the blot was not performed as both at early (30 mins and 60 mins) and late time points (240 mins), we can clearly see reduced levels of Rad53 phosphorylation in Scm3-depleted cells as compared to Cdc20-depleted cells.

Similarly, the new immunoblot for Pds1 in Figure 6E does not give any strong indication that Pds1 degrades earlier in Scm3-depleted cells. (Compare, for example: https://pmc.ncbi.nlm.nih.gov/articles/PMC312708/figure/F5/).

We have now quantified the levels of Pds1 from the graph, and the image below (Figure S1) shows that Pds1 levels are low as it is degraded faster in the absence of Scm3. This is also supported by the presence of anaphase cells at later time points, suggesting that due to the degradation of Pds1, cells bypass the metaphase arrest due to MMS and proceed to anaphase with ploidy problems.

(2) The ChIP-seq experiment remains largely inconclusive regarding whether there is increased binding of Scm3 to chromatin after MMS treatment and whether there is enrichment at fragile sites. The ChIP profiles from untreated and MMS-treated cells look highly similar, and it remains unclear whether differences in peak height between control and MMS are inter-experiment variability or MMS-induced changes. While a few Scm3 peaks overlap with fragile sites based on the new data you report, this is the case both in untreated and in MMS-treated cells. There is no clear increase after MMS treatment and in fact, some sites lose Scm3 enrichment after MMS treatment.

Based on the ChIP-seq data, we detected 33 Scm3 peaks in untreated cells and approximately three times more (90 peaks) in MMS-treated cells. This indicates a potential increase in Scm3 chromatin association following MMS treatment. Previous studies have shown that Scm3 is not restricted to centromeres (Xiao et al., 2011; Wisniewski et al., 2014), and cells lacking Scm3 exhibit partial S-phase arrest; this could be the reason why Scm3 binds to a few fragile sites even without DNA damage. Notably, a greater number of Scm3 peaks are observed at fragile sites after MMS exposure; since untreated cells themselves show Scm3 binding to a few fragile sites, the difference is not as apparent. However, this does not change the fact that Scm3 interacts with fragile sites, and rDNA apart from centromeric regions. The presence of Scm3 at fragile sites, even in the absence of drug treatment, and its increased binding after MMS exposure, highlights the potential importance of Scm3 in maintaining genome stability in these vulnerable regions.